# The influence of atmospheric model resolution in a climate model-forced ice sheet simulation

Marcus Lofverstrom[1] and Johan Liakka[2]

[1]National Center for Atmospheric Research, 3090 Center Green Dr., 80301, Boulder, CO, USA
[2]Nansen Environmental and Remote Sensing Center, Bjerknes Centre for Climate Research

*Correspondence to:* Marcus Lofverstrom (marcusl@ucar.edu)

**Abstract.** Coupled climate–ice-sheet simulations have been growing in popularity in recent years. Experiments of this type are however challenging as ice sheets evolve over multi-millennial time scales, which is beyond the practical integration limit of most Earth-system models. A common method to increase model throughput is to trade resolution for computational efficiency (compromises accuracy for speed). Here we analyze how the resolution of an atmospheric general circulation model (AGCM) influences the simulation quality in a standalone ice-sheet model. Four identical AGCM simulations of the Last Glacial Maximum (LGM) were run at different horizontal resolutions: T85 ($1.4°$), T42 ($2.8°$), T31 ($3.8°$), and T21 ($5.6°$). These simulations were subsequently used as forcing of an ice-sheet model. While the T85 climate forcing reproduces the LGM ice sheets to a high accuracy, the intermediate resolution cases (T42 and T31) fail to build the Eurasian Ice Sheet. The T21 case fails in both Eurasia and North America. Sensitivity experiments using different surface mass balance parameterizations improve the simulations of the Eurasian ice-sheet in the T42 case, but the compromise is a substantial ice buildup in Siberia. The T31 and T21 cases are not improving in the same way in Eurasia, though the latter simulates the continent-wide Laurentide Ice Sheet in North America. The difficulty to reproduce the LGM ice sheets in the T21 case is in broad agreement with previous studies using low-resolution atmospheric models, and is caused by a substantial deterioration of the atmospheric climate between the T31 and T21 resolutions. It is speculated that this deficiency may demonstrate a fundamental problem using low-resolution atmospheric models in these types of experiments.

## 1 Introduction

Experiments with coupled climate–ice-sheet models have become increasingly popular in recent years, much thanks to coordinated international modeling initiatives such as the "Ice Sheet Model Intercomparison Project" (ISMIP6) (Nowicki et al., 2016) and the "Pliocene Ice Sheet Modelling Intercomparison Project" (PLISMIP) (Dolan et al., 2012). These types of experiments are however challenging, as ice sheets have a high thermal inertia that makes their response time greater than almost all other components of the climate system—the time scale depends on the application but typically ranges from $10^3$ to $10^5$ years. Simulations of this length are beyond the practical integration limit of most Earth-system models, and a number of techniques to increase the model throughput have therefore been devised. Some of the more popular approaches for simulating ice sheets over glacial time scales include:

(i) Force a standalone ice-sheet model with a transient climate record obtained by interpolating between the climate extremes over the period of interest (often simulations of the pre-industrial and the Last Glacial Maximum; PI and LGM, respectively). The interpolation weights are typically derived from oxygen isotope ratios in Greenland and Antarctic ice cores (e.g. Charbit et al., 2007; Fyke et al., 2014).

5    (ii) Use an asynchronous coupling between an ice-sheet model and a general circulation model (GCM). The ice-sheet model, which is computationally cheaper than the GCM, is then run multiple years between each update of the model climate (e.g. Liakka et al., 2011; Liakka, 2012; Herrington and Poulsen, 2012; Löfverström et al., 2015).

(iii) Utilize a computationally efficient intermediate complexity model (EMIC) that can be run transiently over glacial time scales (e.g. Roe and Lindzen, 2001; Calov et al., 2005; Bonelli et al., 2009; Ganopolski et al., 2010; Beghin et al., 2014).

10    Although no attempt is made here to assess how these different approaches compare to one another, we conclude that they all rely on a number of assumptions and simplifications that potentially can influence the results. For example: (i) assumes that the glacial climate evolved as a linear combination of the PI and LGM states, which is at odds with both modeling and proxy-data evidence of highly nonlinear circulation changes over the last glacial period (e.g., Jackson, 2000; Zhang et al., 2014; Lora et al., 2016; Pausata and Löfverström, 2015; Löfverström et al., 2016, 2014; Löfverström and Lora, 2017); (ii) accelerating the 15 ice-sheet component introduces abrupt changes in the GCM boundary conditions, which may force the model climate into an unphysical state at the beginning of each (GCM) run segment; (iii) simplified models often rely on statistical dynamics/physics, where almost all interactions are prescribed or represented by first-order linear assumptions.

In addition, one issue that has received little attention in the literature is what role the atmospheric grid resolution—the horizontal mesh on which the model equations are discretized—plays in coupled climate–ice-sheet experiments. Simplified 20 circulation models often utilize coarse horizontal grids for computational efficiency. For example, the atmospheric component of CLIMBER-2 has a horizontal resolution of approximately $10° \times 51°$ (Petoukhov et al., 2000), LOVECLIM runs on a $5.6° \times 5.6°$ resolution grid (Goosse et al., 2010), and FAMOUS on a $5° \times 7.5°$ grid (Smith et al., 2008). These are to be compared with the nominal $1° \times 1°$ resolution of many modern GCMs (e.g. Flato et al., 2013).

Although a higher resolution is not automatically synonymous with a better model, it generally means that smaller scale 25 phenomena can be resolved, which in turn reduces the need for explicit (parameterized) diffusion. Note that diffusion is not only influencing (damping) horizontal motions, but it can also impact vertical transport (Polvani et al., 2004). Several studies have shown that the numerical convergence breaks down somewhere between the T31 ($3.8°$) and T21 ($5.6°$) resolutions in an atmospheric GCM, which (presumably in part due to an increased diffusion rate; Magnusdottir and Haynes, 1999) degrades the representation of even the largest scale atmospheric phenomena, such as jet streams and planetary waves (Polvani et al., 30 2004; Magnusdottir and Haynes, 1999; Dong and Valdes, 2000; Löfverström et al., 2016). This resolution limit appears to be an inherent property of the model dynamics, and thus largely independent of model physics; e.g. Polvani et al. (2004) and Löfverström et al. (2016) found a similar limit using a dry primitive equation model (no model physics), and a comprehensive atmospheric circulation model (fairly sophisticated physics), respectively.

Motivated by the discussion above, the objective of this study is to illustrate that the atmospheric model resolution can have a strong influence on the ice development in climate-model-forced ice sheet experiments. In order to isolate the influence of the atmospheric model resolution we resort to a simplified experiment design (see Section 2) and run an ice sheet model to equilibrium (starting from an ice-free state), using atmospheric forcing data from four identical LGM simulations run at progressively coarser horizontal grids: T85, T42, T31, and T21 (Table 1). This modeling approach takes several steps away from reality, and the study is therefore perhaps best viewed in an abstract light. For example, by prescribing perpetual LGM conditions we ignore the low-frequency, multi-millennial variations in insolation, greenhouse gas concentrations, and atmosphere and ocean circulation that are typically associated with glacial cycles. Moreover, the presence of LGM ice sheets in the atmospheric simulations primes both northwestern Eurasia and northern North America to be susceptible to ice formation. However, in this context this may be considered an asset, as all ice-sheet model experiments theoretically should have a similar bias towards ice formation in the "correct" areas. Also, running the ice-sheet model to equilibrium may seem excessive (it is doubtful that the LGM was an equilibrium state), but it ensures a more objective comparison of the different experiments than is offered by an arbitrarily chosen integration limit.

These shortcomings aside, the ice-sheet model run with the T85 climate forcing manages to reproduce the LGM reconstruction to a high accuracy. The intermediate resolution cases (T42 and T31), on the other hand, fail to reproduce the Eurasian Ice Sheet, and the T21 case fails in both continents. These results suggest that a "sufficiently" high atmospheric resolution may be required to ensure the quality of (coupled) climate–ice-sheet model experiments.

The models and experiment design are presented in Section 2, the results from the atmospheric model and the ice-sheet model are described in Sections 3 and 4, followed by a more general discussion in Section 5.

## 2  Models and experiments

### 2.1  Ice sheet model

We use the three-dimensional ice-sheet model SICOPOLIS (SImulation COde for POLythermal Ice Sheets, version 3.1), run at a 80 km resolution grid that covers most of the Northern Hemisphere. The model treats ice as an incompressible, viscous, and heat-conducting fluid (Greve, 1997), using the shallow-ice approximation (Hutter, 1983) subjected to Glen's flow law (with stress exponent $n = 3$) (e.g. Van der Veen, 2013). A Weertman-type sliding scheme is also applied (Weertman, 1964).

We run the model in the so-called "cold-ice mode", which means that temperatures exceeding the pressure melting point are artificially reset to the pressure melting temperature. The global sea level is lowered by 120 m to reflect LGM conditions, and marine ice is allowed to form where the bathymetry is less than 500 m, otherwise instantaneous calving is applied. The geothermal heat flux is set to a constant global value of $55\,\mathrm{mW\,m^{-2}}$, and the bedrock relaxes toward isostatic equilibrium with a timescale of 3 kyrs, assuming a local lithosphere and relaxing asthenosphere (Greve and Blatter, 2009). All simulations started from ice-free conditions (interpolation of atmospheric fields is described in Section 2.3) and were run for 150,000 years to ensure an objective comparison of the ice sheets' steady state extent.

## 2.2 Ablation parameterizations

SICOPOLIS uses the positive-degree-day (PDD) method to parameterize ablation. The annual melt-potential is estimated from the integrated sum of positive temperatures each year (Braithwaite and Olesen, 1989; Reeh, 1991), assuming that the daily temperatures are normally distributed about the monthly mean value (Calov and Greve, 2005).

Following Charbit et al. (2013), we test the sensitivity of the surface-mass balance scheme using three different PDD-based ablation models: the default parameterizations in SICOPOLIS (based on Reeh, 1991), plus the ones presented in Fausto et al. (2009b), and Tarasov and Peltier (2002) (henceforth referred to as SICOdef, FST09, and TP02, respectively).

These parameterizations use different methods for calculating the degree-day factors for snow and ice ($\beta_{snow}$ and $\beta_{ice}$, respectively), refreezing fraction of melt water ($P_{max}$), and standard deviation (day-to-day variability) of temperature ($\sigma$). All these parameters are set to numerical constants in SICOdef ($\beta_{snow} = 3$ mm day$^{-1}$ K$^{-1}$, $\beta_{ice} = 12$ mm day$^{-1}$ K$^{-1}$, $P_{max} = 0.6$ and $\sigma = 5°$C), while they take on slightly more elaborate expressions in the other parameterizations (see below).

### 2.2.1 The FST09 model

The standard deviation of daily temperature ($\sigma$) is here assumed to change with elevation at a rate of $1.2224°$C km$^{-1}$, starting from $\sigma = 1.574°$C at sea level ($\sigma \approx 4°$C at 2000 m elevation). A similar elevation dependence is also applied to $P_{max}$ to account for the increasing probability of melt water refreezing at higher elevation. No refreezing of melt water ($P_{max} = 0$) is assumed below 800 m, and total refreezing ($P_{max} = 1$) above 2000 m.

In addition, the FST09 model uses a temperature dependent degree-day factor for ice that varies from $\beta_{ice} = 7$ mm day$^{-1}$ K$^{-1}$ for warm boreal summer (June-July-August; JJA) conditions ($\geq 10°$C), to $\beta_{ice} = 15$ mm day$^{-1}$ K$^{-1}$ for cold summer temperatures ($\leq -1°$C). A cubic change is applied for intermediate temperatures. The degree-day factor for snow is a constant with the same numerical value as in SICOdef ($\beta_{snow} = 3$ mm day$^{-1}$ K$^{-1}$).

### 2.2.2 The TP02 model

The TP02 model uses a similar temperature-dependent parameterization of $\beta_{ice}$ as in FST09, but with bounds: $\beta_{ice} = 8.3$ mm day$^{-1}$ K$^{-1}$, and $\beta_{ice} = 17.22$ mm day$^{-1}$ K$^{-1}$ for warm ($\geq 10°$C) and cold ($\leq -1°$C) summer (JJA) temperatures, respectively. A similar parameterization is also applied to $\beta_{snow}$ that varies between $\beta_{snow} = 4.3$ mm day$^{-1}$ K$^{-1}$ and $\beta_{snow} = 2.65$ mm day$^{-1}$ K$^{-1}$, respectively. The standard deviation of temperature is set to a constant value of $\sigma = 5.2°$C. The refreezing scheme is also more comprehensive (based on Pfeffer et al., 1991; Janssens and Huybrechts, 2000), including both thermodynamics (latent heat release due to refreezing) and pore trapping components.

## 2.3 Climate evolution in SICOPOLIS

The surface temperature and precipitation (over the evolving ice sheets) are calculated using the method described in Liakka et al. (2016), which in turn is based on the general methodology outlined in Charbit et al. (2002, 2007). While the temperature decreases linearly with height at a fixed lapse rate $\gamma$ ($= -6.5 \times 10^{-3}$ K m$^{-1}$; the "standard" atmospheric lapse rate is assumed

as the actual value over the LGM ice sheets is unknown; see Section 5 for a motivation of this choice), the precipitation amount changes exponentially as a function of temperature; see Eqs. 1 and 2 in Liakka et al. (2016). The distribution of liquid and solid precipitation is also assumed to vary with temperature: 100% solid precipitation falls if the monthly mean surface air temperature is below $-10^\circ$C, and 100% liquid if it is higher than $7^\circ$C. The distribution changes linearly for intermediate tem-

peratures (Marsiat, 1994). The surface mass balance (SMB) is calculated from the climatological monthly-mean temperature and precipitation fields, which are (bilinearly) interpolated from the atmospheric (LGM) simulations, using the above lapse rate to correct for elevation biases due to different grids and horizontal resolutions.

## 2.4   Atmosphere model

The atmospheric climate forcing is produced with the Community Atmospheric Model version 3 (CAM3) (Collins et al., 2004,

2006b), using four different spectral (horizontal) resolutions: T85, T42, T31, and T21, corresponding to an approximate grid spacing of $1.4^\circ$, $2.8^\circ$, $3.8^\circ$, and $5.6^\circ$, respectively (Table 1). The model uses identical parameterizations (same equations) at all horizontal resolutions (Collins et al., 2004, 2006b), but the climate is tuned by varying twelve parameters governing the representation of clouds and precipitation (convective and stratiform), biharmonic diffusion, and integration time step in order to satisfy the Courant-Friedrichs-Lewy (CFL) condition; some of the resolution-dependent parameter settings are presented

in Table. 1 (see Collins et al., 2004, for a complete model description). Note that the model physics is represented in grid space, while the dynamics is discretized in spectral space. The effective diffusion rate is thus scale dependent and modulated by (horizontal) wave number in the vorticity and divergence equations (Collins et al., 2004).

The planetary boundary conditions are set to reflect LGM conditions, including the orbital parameters and greenhouse gas concentrations outlined by the Paleoclimate Modeling Intercomparision Project (PMIP) (e.g. Kageyama et al., 2017), the ice

sheet reconstruction presented by Kleman et al. (2013) (raised to the height of the ICE-5G reconstruction (Peltier, 2004) to encourage ice formation in the "correct" areas in SICOPOLIS), and prescribed monthly varying sea-surface conditions (LGM sea-surface temperature and sea-ice extent) from Otto-Bliesner et al. (2006). Motivated by the official PMIP boundary conditions (e.g. Kageyama et al., 2017), the vegetation cover in non-glaciated areas is prescribed as the modern distribution.

The grid-resolved boundary conditions were spectrally interpolated (using the same spectral transforms as in the atmospheric

model) from the T85 grid to the coarser resolutions in order to ensure an identical setup. However, the spectral smoothing in the interpolation process lowers the resolved topography (including the ice sheets) on the coarser resolution grids. The interior of the Laurentide Ice Sheet is at most a few hundred meters lower in the T21 case, but somewhat larger differences (500 to 1000 m; Fig. S1) are found over the Eurasian Ice Sheet as this is of smaller horizontal extent, and thus less well defined at lower resolutions. All simulations were run for 12 years, from which monthly climatologies were created over the last 10 years. The

short integration length is motivated by the idealized experiment design (see Section 1) and the prescribed (perpetual LGM) sea-surface conditions that help dampen atmospheric interannual variability. A longer sampling rate may alter details in the climatologies, but is not expected to change the first order conclusions from the study.

## 3 Climate forcing at different horizontal resolutions

In order to understand how the model climate responds to the horizontal resolution, we begin by comparing fields that are strongly related to model dynamics/physics, using the T85 case as a benchmark for the comparison. Figure 1a-d shows the 500 hPa eddy stream function (proportional to high- and low-pressure regions) and zonal wind in boreal summer (JJA). In agreement with previous studies (e.g. Dong and Valdes, 2000; Polvani et al., 2004; Magnusdottir and Haynes, 1999; Löfverström et al., 2016), the large scale atmospheric dynamics is well captured at the T42 and T31 resolutions—the circulation patterns have similar amplitude and spatial distribution as the T85 case—but it deteriorates substantially at the T21 resolution. A somewhat more gradual change is seen in the (vertically integrated) cloud cover (Fig. 1e-h), which is strongly controlled by the physics parameterization. The cloud cover changes from about 50% over the ice sheets in the T85 case, to almost 100% in the T21 case.

Related to this discussion, Figs. 2 and 3 show the surface temperature and precipitation climatologies that are used as forcing of the ice-sheet model; the full fields are presented in the left columns (panels a-d), and the difference with respect to the T85 case are shown on the right (panels e-g). We focus on the surface temperature in boreal summer as this is the primary ablation season, but the cumulative sum of precipitation over the year (total annual), as ice can form in all seasons in regions with a positive surface mass balance.

The JJA surface temperature is to first order similar in the two intermediate resolution cases (T42 and T31; Figs. 2e,f), featuring a localized warming with respect to the T85 simulation over the northern parts of the Laurentide Ice Sheet, the interior of the Greenland Ice Sheet, and most of the Eurasian Ice Sheet. This is partially a response to the lowering (smoothing) of the resolved topography on the coarser grids, but the majority of the warming is related to changes in the surface energy balance induced by the increased cloudiness (see discussion in Section 5). The largest differences in precipitation are found in the midlatitude storm tracks that shift equatorward relative to the T85 case, especially in the North Atlantic (panels 3). A similar resolution-induced storm-track shift has been found in several atmospheric models (Guemas and Codron, 2011; Hourdin et al., 2012; Demory et al., 2014), and thus appears to be fairly robust and largely independent of grid type and physics parameterizations.

The T21 case shows a fairly different response with a considerable warming over most of the world's topography (including the ice sheets; Fig. 2g). This is partly a response to the lower mean-height of the resolved topography (smoothing from the interpolation process), but also from a general degradation of the model climate and an enhanced downwelling of longwave radiation due to the increased cloudiness (Fig. 1; see further discussion in Section 5). The midlatitude precipitation field is also considerably altered with respect to the T85 case, with substantially lower precipitation in the eastern parts of the midlatitude storm tracks and thus over the southwestern parts of the ice sheets (Fig. 3). Note that this is presumably a response to the model's inability to resolve planetary waves (and hence individual cyclones) at coarse horizontal resolutions (Fig. 1; see also Polvani et al., 2004; Magnusdottir and Haynes, 1999; Guemas and Codron, 2011; Hourdin et al., 2012; Löfverström et al., 2016).

## 4 Ice sheet model results

The left column in Fig. 4 shows the equilibrium ice-sheet extent when using the default SMB parameterization in SICOPOLIS (SICOdef). The ice sheets forming under the high resolution atmospheric climatology (T85; panel 4a) are in close resemblance with the target extent (indicated by solid contours; Kleman et al., 2013), with only slightly too much ice extending in western Canada and along the Siberian Arctic coast.

The ice sheets forced by the intermediate resolution climatologies (T42 and T31; panels 4b,c) adequately reproduce the North American ice sheet, but they they fail to build the Eurasian counterpart in agreement with the reconstruction. This one-sided mismatch can be understood from the atmospheric climatologies described in Section 3. The warm summer temperature over the southwestern parts of the Eurasian Ice Sheet (Figs. 2e,f) is the main reason for why ice is not forming in this region. Note that although there is a relatively small reduction of precipitation with respect to the T85 case (the interior of Scandinavia is actually showing larger values than the T85 case), the warm surface temperatures are by far the most pronounced feature over the Eurasian Ice Sheet (cf. Figs 2 and 3; see discussion in Section 5). These results are in broad agreement with Abe-Ouchi et al. (2013), who showed that the Eurasian Ice Sheet is more sensitive to temperature changes than the North American counterpart. The relatively small temperature change over the Eurasian Ice Sheet is thus sufficiently strong to influence the ice sheet expansion there. The warm signal in northwestern North America (Figs. 2e,f) is located in a relatively cold region with a short ablation season, and therefore has a comparatively smaller influence on the local ice sheet evolution.

The T21 case, on the other hand, struggles to reproduce the LGM ice sheets in both continents. Although ice forms in North America, it fails to build the continent-wide Laurentide Ice Sheet and instead forms two distinct ice sheets—a smaller eastern and a larger western dome—separated by a wide gap in the region around Hudson Bay. This response bears some structural similarity to the low-resolution model results shown in Beghin et al. (2014) and Charbit et al. (2013), and also the pre-LGM ice sheets in Calov et al. (2005), and Bonelli et al. (2009). Similar to the T42 and T31 cases, the T21 climate forcing is too warm (and presumably too dry) over the southwestern parts of the Eurasian Ice Sheet area to reproduce the LGM ice sheet reconstruction.

The sensitivity experiments with different SMB parameterizations in SICOPOLIS are presented in Fig. 4e-l. The middle row (panels 4e,f,g,h) uses the FST09 ablation model, and the bottom row (panels 4i,j,k,l) the ablation model described in TP02. Both these alternative SMB parameterizations help improve the Eurasian ice extent in the T42 case (Figs. 4f,j), though at the price of a fairly substantial ice buildup in northern Siberia and Beringia (particularly pronounced in Fig. 4f), which are areas that were largely ice free at the LGM (Svendsen et al., 2004; Kleman et al., 2013; Löfverström and Liakka, 2016). A broadly similar buildup in these regions is also seen in the T85 case when using these SMB parameterizations.

These alternative SMB parameterizations are not improving the ice-sheet simulations in Eurasia when using the lower resolution climatologies (T31 and T21; Fig. 4g,h,k,l), but they help the formation of a continent-wide Laurentide Ice Sheet in the T21 case (Fig. 4k,l).

## 5 Discussion and conclusions

The results presented in this paper attempt to illustrate, albeit in a highly qualitative way, the influence of atmospheric resolution on climate-forced ice-sheet-model simulations. By adopting a simplified modeling approach we can effectively isolate the resolution dependence of the atmospheric model, and by prescribing LGM boundary conditions (sea-surface conditions and continental ice sheets), the ice formation is primed to occur in the "correct" areas in the subsequent ice-sheet model experiments. This methodology appears to work well when using the high resolution atmospheric climatology (T85; see also Liakka et al., 2016), but is less successful when using the climatologies from the lower resolution simulations (T42, T31, and T21; Fig. 4). The analysis shows that both the simulated surface temperature (Fig. 2) and precipitation (Fig. 3) fields are changing in ways that hinder ice from forming in the "desired" areas at the lower resolutions. The precipitation changes are however found to be secondary, hence we devote the first part of this discussion to exploring the origin of the warmer surface temperatures.

There are primarily two explanations for why the surface temperatures increasing at lower horizontal resolutions: (i) lapse-rate effects due to differences in resolved topography; and (ii) changes in the simulated climate that are conducive for warm surface temperatures over the LGM ice sheets. We discuss these processes in the next two paragraphs:

(i) Moving to a coarser horizontal resolution typically results in a lapse-rate induced surface warming, as the resolved topography is both lower and smoother as a result of the increased grid spacing. In this study we employed the modern global-average lapse rate of $6.5\,°\mathrm{C\,km^{-1}}$ for vertical interpolation/extrapolation. This is about 1 to $2\,°\mathrm{C\,km^{-1}}$ higher than observations over the Greenland Ice Sheet in boreal summer (Fig. S2; Fausto et al., 2009a), but is motivated by the generally drier conditions in glacial climates that shift the lapse rate towards higher values (Clausius-Clapeyron scaling; LGM simulations typically feature a global annual cooling of 4 to $6\,°\mathrm{C}$ relative to pre-industrial; e.g. Braconnot et al., 2007)—Loomis et al. (2017) showed that the tropical atmospheric lapse rate may have increased from about $5.8\,°\mathrm{C\,km^{-1}}$ in the modern climate, to $6.7\,°\mathrm{C\,km^{-1}}$ at the LGM. The elevation difference in the interior of the Laurentide ice sheet is around $200\,\mathrm{m}$ between the T85 and T21 cases (Fig. S1), hence the lapse-rate effect is only accounting for 5-10% of the local warming signal in Fig. 2. The lapse-rate effect is however more important on the ice sheet edges and in Eurasia (accounting for 30 to 50% of the warming signal), where the difference in topography is larger.

(ii) The majority of the temperature difference in Fig. 2 is induced by changes in the atmospheric circulation. The stationary planetary waves are considerably weaker in the T21 case (Fig. 1), resulting in reduced cold-air advection over the Laurentide Ice Sheet (Fig. S3). The total cloudiness is at the same time significantly higher (Fig. 1). While clouds help regulate the amount of downwelling shortwave radiation at the surface, upper level ice-clouds increase the re-emission of longwave radiation back to the surface. Changes in cloudiness are found to increase the surface radiative heating effect ($\mathrm{SW_{net}} + \mathrm{LW_{down}}$) by 10 to $30\,\mathrm{W\,m^{-2}}$ over the LGM ice sheets (Fig. S4).

As a result, while the T42 and T31 cases struggle to build ice in Eurasia, the T21 experiment also fails to build the continent-wide Laurentide Ice Sheet in North America (when using the default SMB parameterization; SICOdef). Instead it builds two spatially disconnected ice sheets, with a larger dome on the western side of the continent (Fig. 4d). Several coupled climate–ice-sheet experiments with a low-resolution atmospheric model have shown qualitatively similar results, see e.g.: Calov et al.

(2005); Charbit et al. (2013); Beghin et al. (2014). The common denominator for these studies is that they all used CLIMBER-2 to produce the atmospheric forcing fields. We stress that it is not our intention to single out this particular model, but it appears to suffer from similar deficiencies as our T21 case and may therefore help us understand some of these results. In the aforementioned papers the ice sheet tends to be limited to the western/northwestern side of the North American continent

(e.g. Charbit et al., 2013; Beghin et al., 2014), little or no ice is established in western Eurasia (e.g. Calov et al., 2005; Charbit et al., 2013; Beghin et al., 2014), and attempts to remedy these shortcomings typically result in substantial ice formation in Siberia and Alaska (see Charbit et al., 2013, who tested the sensitivity of the same PDD-based SMB parameterizations as were used in this study). These results appear to be largely independent of both the choice of ice-sheet model (the above studies used SICOPOLIS and GRISLI), and the complexity of the SMB parameterization (Charbit et al., 2013; Bauer and Ganopolski,

2017). Although it is not completely fair to compare CLIMBER-2 to a low resolution version of CAM3 (the complexity and general purpose of these models are extremely different), it is possible that these similarities demonstrate a fundamental problem with low-resolution climate models that transcends model complexity.

One piece of information that is rarely mentioned in the literature is that most Earth-system models are tuned to reproduce the climate of the instrument era ($\sim$1850 to present). These models are of course valuable tools for exploring other time periods

as well, but it generally means that inter-model discrepancies tend to increase under more extreme forcing scenarios, e.g. glacial conditions (e.g. Braconnot et al., 2007). The results presented here suggest that the model spread may be further exacerbated by differences in horizontal resolution.

The atmosphere model used here has been tuned and extensively tested at the T85, T42, and T31 grids (e.g. Collins et al., 2006a; Yeager et al., 2006). However, the T21 resolution only has "functional support", which means that boundary conditions

are provided but the model climate has not been tuned to the same standard as the other resolutions. This is probably at least a partial explanation for the apparent degradation of the model climate, though it is possible that this manifests a more general breakdown of the numerical convergence that has been identified in previous modeling studies (e.g. Polvani et al., 2004; Magnusdottir and Haynes, 1999; Dong and Valdes, 2000). Some evidence of this is seen in Fig. 1: while the model physics shows a fairly gradual change between the T85 and T21 resolutions (Fig. 1e-h)—including a generally increased cloudiness and

an equatorward migration of the mid-latitude precipitation field (a similar response to horizontal resolution has been identified in studies of the modern climate; e.g.: Hack et al., 2006; Guemas and Codron, 2011; Hourdin et al., 2012; Demory et al., 2014)—fields more strongly associated with the model dynamics retain much of their amplitude and general structure at the T31 resolution, but deteriorate significantly when going to T21. What manifests an acceptable simulation quality is subjective and highly dependent on application. However, since ice sheets are sensitive to feedback loops triggered by deviations from

"expected" climate conditions (both in terms of mean state and variability), coupled climate–ice-sheet simulations generally require a higher simulation quality than more traditional modeling experiments.

On the other hand, resorting to a lower horizontal resolution can both increase the model throughput (number of simulated years per day), and reduce the simulation cost (CPU-hours per simulated year; e.g. Yeager et al., 2006). As shown in Table 1, simulating one model year on the T85 resolution requires around $21\times$ as many numerical operations as one model year on the

T31 grid, and $48\times$ as many operations for the same integration length on the T21 grid. This encapsulates the challenges of

coupled climate–ice-sheet experiments, as it is common to trade resolution ("accuracy") for computational efficiency ("speed") in order to run transient simulations over glacial timescales.

Lastly, it is possible that some of the shortcomings discussed here—e.g. the lack of ice forming in western Eurasia in the T42 and T31 cases, and in east-central North America in the T21 case—-may be due to the simplified experiment design and selected parameter values (some hints of this is seen in Fig. S2). However, it is important to stress that the ice evolution is ultimately controlled by the quality of the atmospheric forcing data, which we can show is strongly compromised at sufficiently coarse horizontal grids. Based on these results we conclude that a lower practical resolution bound for traditional climate-model experiments is likely to be somewhere around T31, and possibly somewhat higher (nominal T42 or even T85 resolution) for coupled climate–ice-sheet simulations.

*Competing interests.* The authors declare that they have no competing interests.

*Acknowledgements.* We thank the editor Thomas Mölg, two anonymous reviewers, and Irina Rogozhina and Raymond Sellevold for critically evaluating this manuscript. We acknowledge B. Otto-Bliesner and J. Kleman and their collaborators for producing and making publicly available the CCSM3 LGM simulation and LGM ice-sheet reconstruction that were used as basis for our experiments. The AGCM simulations were performed on resources provided by the Swedish National Infrastructure for Computing (SNIC) at the National Supercomputing Center (NSC) that is financially supported by Swedish Research Council (Vetenskapsrådet; VR). The ice-sheet model simulations were carried out on resources provided by LOEWE Frankfurt Centre for Scientific Computing (LOEWE-CSC). This work was financially supported by the National Science Foundation (NSF) and the US Department of Energy (DOE).

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

**Table 1.** Resolution specific details. The top two rows show the horizontal resolution in degrees [°] and in number of grid cells [lat×lon], respectively. The run cost (third row) is normalized with respect to the T21 case and estimates the number of numerical operations required to simulate one model year, based on the grid size and the nominal time step [s] for each resolution (fourth row). The horizontal biharmonic (fourth order) diffusion coefficient is given in units of $10^{15}\,\mathrm{m^4 s^{-1}}$ (bottom row).

|            | T85     | T42    | T31   | T21   |
|------------|---------|--------|-------|-------|
| Resolution | 1.4     | 2.8    | 3.8   | 5.6   |
| Grid size  | 128×256 | 64×128 | 48×96 | 32×64 |
| Run cost   | 48      | 6      | 2.25  | 1     |
| Time step  | 600     | 1200   | 1800  | 1800  |
| Diffusion  | 1       | 10     | 20    | 20    |

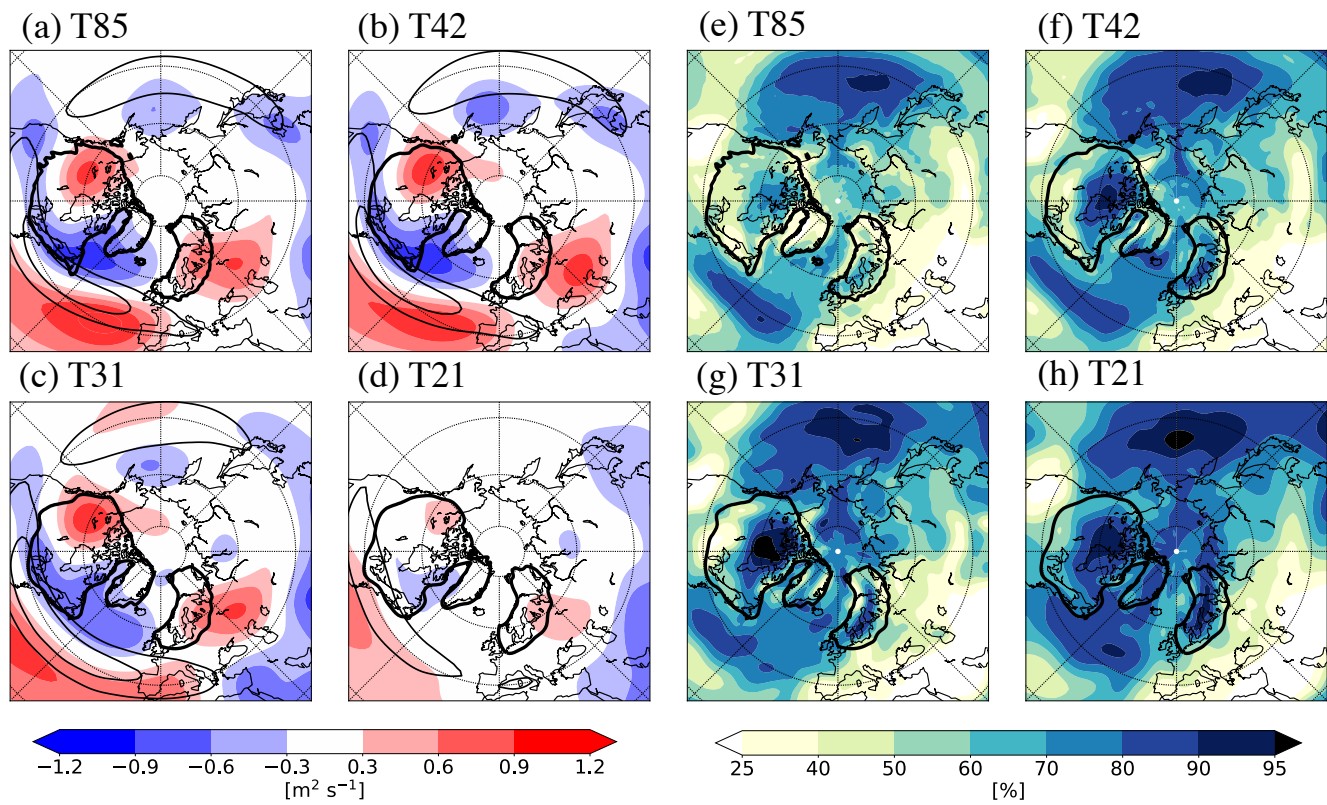

**Figure 1.** (Left) Summer (JJA) 500 hPa eddy streamfunction $[\mathrm{m}^2\mathrm{s}^{-1}]$ (shading; zonal mean removed) and zonal wind $[\mathrm{m\,s}^{-1}]$ (contours; $10\,\mathrm{m\,s}^{-1}$ intervals starting at $20\,\mathrm{m\,s}^{-1}$); (right) vertically integrated (total) cloudiness [%]. The 500 m ice-sheet topography from the LGM reconstruction is indicated by the heavy contours (interpolated to the different horizontal resolutions).

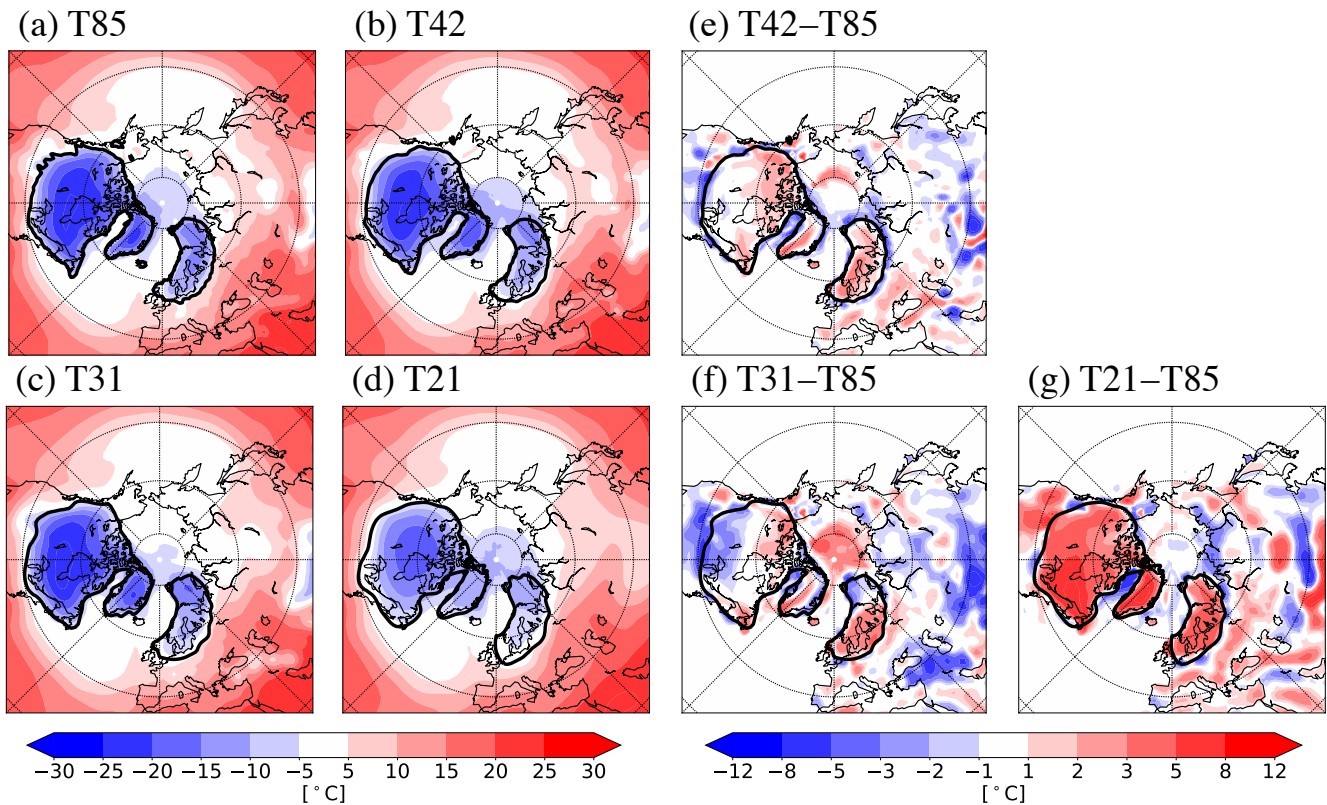

**Figure 2.** Summer (JJA) surface temperature [°C] from the different resolution atmospheric climatologies. The full fields are shown in the left panels (a,b,c,d), and the difference with respect to the T85 case is shown on the right (e,f,g). The 500 m ice-sheet topography from the LGM reconstruction is indicated by the heavy contours (interpolated to the different horizontal resolutions).

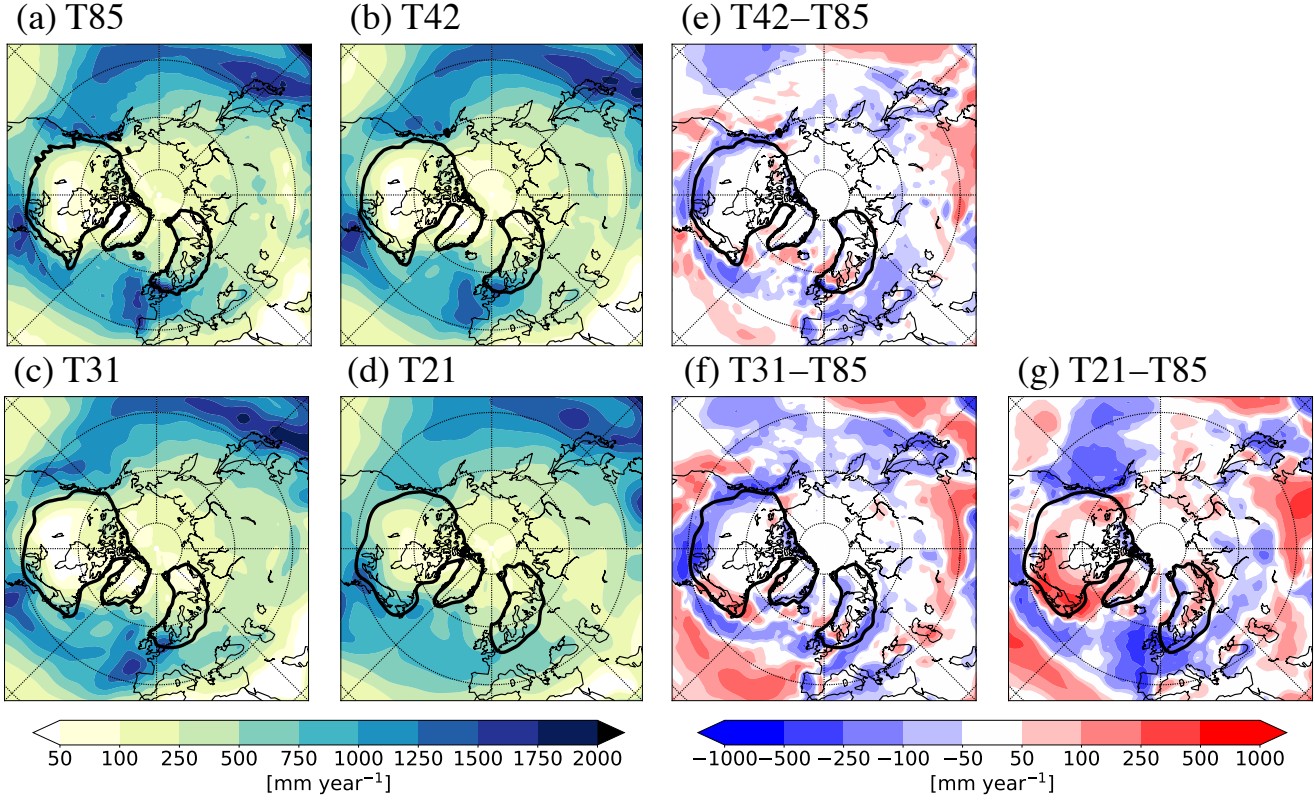

**Figure 3.** Cumulative sum of precipitation (liquid + solid) over the year (total annual amount) [mm year$^{-1}$] from the different resolution atmospheric climatologies. The full fields are shown in the left panels (a,b,c,d), and the difference with respect to the T85 case is shown on the right (e,f,g). The 500 m ice-sheet topography from the LGM reconstruction is indicated by the heavy contours (interpolated to the different horizontal resolutions).

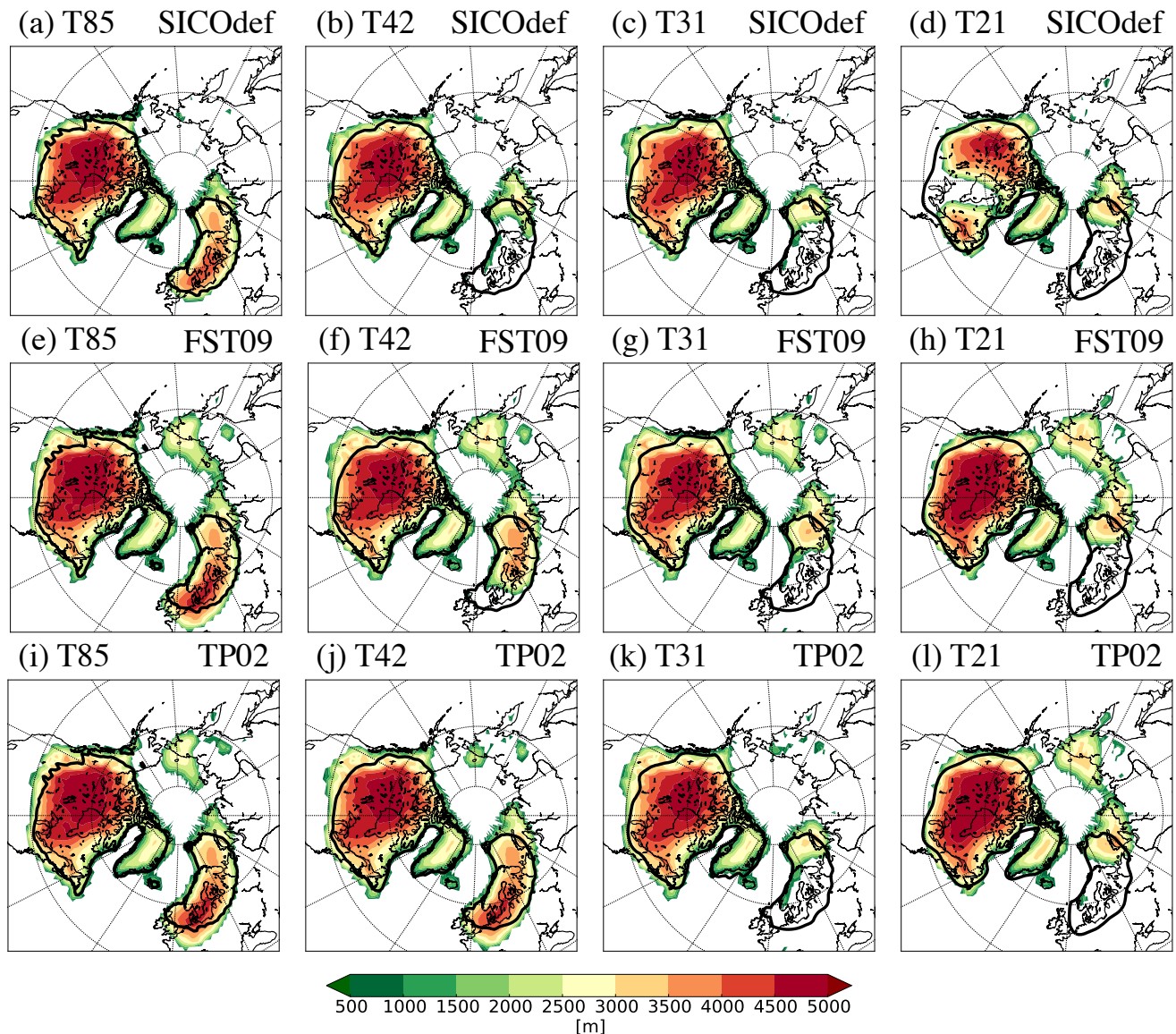

**Figure 4.** Equilibrium ice thickness [m] when using different ablation parameterizations in the surface mass balance scheme: (top) default method in SICOPOLIS; (middle) method by Fausto et al. (2009b); and (bottom) method by Tarasov and Peltier (2002), using the atmospheric climatologies from the (a,e,i) T85; (b,f,j) T42; (c,g,k) T31; and (d,h,l) T21 resolution simulations, respectively. The 500 m ice-sheet topography from the LGM reconstruction is indicated by the heavy contours (interpolated to the different horizontal resolutions).