# Peer review of "The influence of atmospheric model resolution in a climate model-forced ice sheet simulation"

_The Cryosphere, 2017_

## Referee Comment (RC1) · Anonymous Referee #1 · 5 Jan 2018

This short manuscript presents an assessment of the influence of atmospheric model resolution in coupled climate-ice-sheet simulations. It shows that the atmospheric resolution matters enormously for an accurate simulation of the major LGM ice sheets. The manuscript is concise, clearly written, easily readable and presents an important, albeit un-surprising, result. This study can prove to be an easy to read and quick to fall back on article when introducing new and old glaciologists fresh into these kinds of simulations.

My main concern with the manuscript is whether the differences between the different atmospheric resolutions is of dynamical/physical nature, or just a matter of res-

olution and the model topography. The authors argue that the main cause of accurate/inaccurate simulation of the ice sheets is an inaccurate temperature field. Differences in precipitation are rather small. I think that temperature is a very straightforward parameter to model correctly: it largely depends on elevation, which directly depends on the model resolution. Therefore, this conclusion could have been found without doing any of the model simulations presented. How robust are the temperature anomalies shown in Fig. 2? Are these resolution differences also found for other atmospheric models (used for ice-sheet forcing)? Or is it CAM3-specific? The authors should make this finding more convincing by comparing (more) the results to other studies. Currently, in the discussion section, much attention is on the quality of the T21 forcing. I would like to see some more focus, in this section, on the intermodel resolution differences.

In order to warrant publication this concern should be addressed and made less qualitative. Possibly some of the following questions could be addressed in more detail: What do the results of this study say about current studies of coupled atmosphere-ice-sheet models? Have other studies been conducted with inaccurate climate forcing; is this a big issue or not? Have other studies attacked and/or addressed the simple temperature discrepancies; possibly by additional topography (down)scaling techniques, spectral diffusivity, lapse rate corrections? Does the current glaciological community realize that the atmospheric resolution is as important for the results of these types of simulations? How large is the trade-off between "accuracy" and "speed"?

A different approach might be to use different atmospheric climatologies at several model resolutions to make the results more robust, but I understand if this might prove to be beyond the work of this study.

Because I do like the brevity of this manuscript and the concise and to the point explanation of this problem, I do suggest publication after minor revisions. Hopefully the authors can tackle my concerns adequately.

I have no additional minor comments, except for a typo on p6, line 6: considerable->

considerably.

---

## Short Comment (SC1) · 17 Jan 2018

I have major concerns about the experimental design of this study and how the authors' choices affect the main conclusions of their manuscript. One of such choices is the lapse rate correction. A temperature lapse rate correction is used to derive the temperature forcing at the beginning of ice sheet simulations, which are initiated from the present-day ice-free topography (excluding the Greenland Ice Sheet), and to adjust the temperature forcing to the growth of ice masses throughout the simulations. I question both (1) the choice of the lapse rate correction and (2) the assumption of the initial ice-free conditions, which are undermining the very core of this study.

[Figure]

Problem (1): While the authors discuss potential impacts of PDD parameters, they do not question their choice of the temperature lapse rate (6.5C/km). Fausto et al. (2009) measured lapse rates as low as 4.7 and 4.6C/km during the June and July months, respectively (the months of the strongest ablation) on the Greenland ice sheet. This is nearly 2C/km below the value that the authors use throughout the year. The temperature lapse rate is used by the authors to correct the model-based air (or surface) temperatures for the difference between their LGM topography in the atmospheric simulations and ice-free present-day topography assumed at the beginning of their ice sheet simulations. This initial temperature forcing is crucial to the development of an ice sheet: If it is excessively above 0ïĆřC during the summer period, an ice sheet will not build. A difference of ∼2C/km would reduce the corrections of the near-surface temperatures over the areas covered by the former British-Irish and Fennoscandian ice sheets by 3-4C during the months that matter most for the ice sheet surface mass balance, but would not impact Arctic Siberia. Quoting from the study of Löfverström and Liakka, page 1, line 9: "Sensitivity experiments using different surface mass balance parameterizations improve the simulations of the Eurasian ice sheet in the T42 case, but the compromise is a substantial buildup in Siberia". This compromise does not have to be made: The choice of a lower lapse rate correction will trigger a buildup of ice masses over the British Isles and Fennoscandia when using the T42 & T31 climate datasets, reasonable choices of PDD parameters and a higher ice sheet model resolution (see below) but will still keep Arctic Siberia ice free.

Problem (2): A more important question is whether such corrections should be applied at all. While talking about "the influence of atmospheric model resolution in coupled climate-ice sheet simulations" (quoted from the title of the paper), one would rather think about whether the use of lower atmospheric model resolution contaminates the climate state in such a way that it becomes inconsistent with the modeled/prescribed ice sheet geometries (included as a topographic boundary condition in a climate model). The question is not whether this climate forcing can build ice sheets if it's heavily modified using lapse rate corrections but whether it can maintain reasonable

[Figure]

ice sheet geometries when unmodified atmospheric model outputs are used.

While Problem (1) can be easily tackled by performing additional sensitivity experiments, Problem (2) is more challenging to resolve. The authors could test T42, T31 and T21 datasets using ice sheet modeling results from the T85 dataset "reproduced to a high accuracy" (quoted from Löfverström and Liakka, page 1, line 8) to address the question, whether the degradation of the atmospheric model resolution results in ice sheet collapses consistent with their current conclusions. However, their modeled ice sheets in the T85 simulation are 1.5 to 2 km too thick relative to the existing reconstructions of the LGM ice sheet geometries. Even without additional lapse rate corrections (introduced to reconcile the difference between the ice sheets prescribed in the atmospheric simulations and derived from the T85 simulation), removing such thick ice sheets would be a difficult task for the T42, T31 and even T21 data sets. At this point a question arises: Why are the modeled ice sheets so unrealistically thick? I envision several potential causes of such unrealistic model performance: (i) The spin-up of the ice sheet model: Running an ice sheet model to an equilibrium with the LGM climate over 150 thousand years is not in line with the existing evidence. Most of the former ice sheets were short-lived (tens of thousands of years from buildup to decay) (ii) Shallow ice approximation (excluding ice stream dynamics) in combination with excessively low resolution of the ice sheet model (80 km) fails to approximate the rapid ice flow and routing of ice masses towards the ocean. (iii) The isostatic adjustment scheme may cause an exaggerated bending of the bedrock surface under the weight of growing ice sheets (I have not seen the Local Lithosphere and Relaxing Asthenosphere method being used in years).

The overall quality of the study could improve if the authors address problems (1) and (2). It can also benefit from the use of SICOPOLIS v3.3 that includes options for higher resolution, more realistic treatment of ice streams and glacial isostatic adjustment. Finally I strongly suggest that the authors improve their figures. The adopted projection strongly distorts the Arctic region, which is the main focus of the present study.

References Fausto, R. S., Ahlstrøm, A. P., Van As, D., Bøggild, C. E., and Johnsen, S. J.: A new present-day temperature parameterization for Greenland, J. Glaciol., 55, 95–105, 2009.

---

## Short Comment (SC2) · 17 Jan 2018

You mention in the introduction how horizontal diffusion does not only influence horizontal motions, but may also impact vertical transport and convection. I wonder how important resolving convection adequately is for building/removing an ice sheet. It would be informative if you more specifically related resolution dependent dynamics/physics with possible shortcomings in building/removing the ice sheets.

In your method section, you describe that present day non-glaciated areas are prescribed with modern day vegetation cover. Do you think this assumption is valid? Do you think it may have a large influence on the simulations?

[Figure]

You present the smoothed topography as a reason why some areas are warmer when the resolution is lower. After that, areas that are colder with respect to the T85 case are pointed out with no explanation why they may be colder.

The tropical and midlatitude precipitation fields are well covered in the text, but there is not much mentioning of precipitation over the ice sheets. It is also hard to see the difference of this field between different resolutions, since the colors starts at 200 mm/yr. A possible solution would be to make a non-linear color scale to better resolve the low-precipitation areas (such as the ice sheets).

"The ice sheets forming under the high resolution atmospheric climatology (T85; panel 3a) are in close resemblance with the target extent (indicated by solid contours; Kleman et al., 2013). There is essentially only too much ice extending along the Siberian Arctic coast." Why is there too much ice extending there? Because it is very cold, because the precipitation is very high, or maybe a combination of both?

"However, the T21 resolution only has "functional support", which means that boundary conditions are provided but the model climate has not been tuned to the same standard as the other resolutions (the resolution dependent tuning parameters are broadly the same as in the T42 case)." How is the 1850-present climate in this low resolution as it is not tuned? Was there any attempt made to tune it? If no, why not?

---

## Referee Comment (RC2) · Anonymous Referee #2 · 18 Jan 2018

Review of "A note on the influenced of atmospheric resolution in coupled climate—ice-sheet simulations":

The manuscript examines the effect of atmospheric resolution on ice sheet modeling forced with climate model output. The theme is certainly relevant for the emerging research on coupled ice-sheet/climate modeling, both in the context of future climate projections as well as paleo-research, and timely due to launch of international collaborative projects such as ISMIP6. To my knowledge, this topic has not been directly addressed in a systematic way like this before.

The method applied consists in forcing an ice sheet model (SICOPOLIS, using the

[Figure]

Shallow Ice Approximation), with climate output of different resolutions corresponding to the LGM. The ice sheet model is started from zero thickness. PDDs are applied for the surface mass balance calculation.

General comment: I would like to see more analysis of the climate model in addition to see the results of applying it as forcing to the ice sheet model. I'll explain in the following. The outcome of the study, namely the identification of a threshold resolution beyond what the climate simulation quality is compromised, is a very interesting result. For this reason, I would urge the authors to expand on the reasons (that is, physical processes lacking, misrepresented, and/or well-captured in the model at each resolution) for this threshold. In other words, what makes the low-resolution model unable to capture the essence of the LGM climate?

Introduction: The topic is very well introduced/motivated and the literature review is a great piece of work.

Comment on methodology: The method relies in strong assumptions and approximations. The simulation target is to reproduce the reconstructed LGM ice sheets (by Peltier et al.) by building them from zero thickness under a steady-state LGM forcing. In reality, there was a history of building up these ice sheets, so they are not the effect of a constant LGM climate. However, the method seems an efficient reasonable approach for the objective of the manuscript, and seems to work in the identification of a threshold for "minimum required resolution".

Other comments: It is difficult to follow the precipitation discussion due to the choice for the color bar. Polar latitudes have low precipitation rates, please use a suitable color bar, albeit the loss of resolution for the tropical area. I would remove the words "a note" from the title after expanding the manuscript with further climate model analysis. Also, the study does not include "fully" coupled climate—ice-sheet simulations in the sense that the climate model is not influenced by the ice sheet model in any way. The authors probably chose the wording "in coupled (. . .) simulations" in the context of motivation,

but the title can be misleading about the content of the actual study. I would replace the title for e.g. "On the influence of atmospheric resolution on climate-model-forced ice sheet simulations"

---

## Editor Comment (EC1) · T. Mölg (Editor) · 23 Jan 2018

I would like to thank the two anonymous reviewers and Irina Rogozhina and Raymond Sellevold for their comments and critical reading. My following remarks are thought to help in putting together your final responses.

Based on all the comments, there will have to be some expansions to the manuscript, however without having to give up on the general character of this study of being presented in a rather concise format (which I support). I suggest to pay attention to two points in particular.

[Figure]

1) A bit more analyses of the results (as I assumed in my access review). – I would in any case suggest to tackle the lapse rate point by sensitivity analyses, and elaborate a bit more on the atmospheric model differences (causes).

2) More justification on the study design, see in particular the comment by Irina Rogozhina (and partly RC2). – I would like to add to this issue that the paper would also profit from clarifying the term "coupled" in the given context. Your study is clearly "one-way" (or "standalone", as you say in the paper) with regard to the ice sheet model. Most probably, for the general reader "coupled" would imply "interactive". Therefore, (a) be cautious with using "coupled" here (e.g., not in the title, see RC2 as well) and (b) provide a short argument early in the paper why standalone simulations make sense and/or are still state-of-the-art in face of the comments by Irina Rogozhina. RC2 also suggests in this regard you could argue with the objective of your study.

Small things to consider:

(i) Please explain/argue briefly why only 12 years are simulated with the AGCM. For the general reader, this might be unclear since you talk in the introduction about ice sheet response time scales of hundreds to thousands of years.

(ii) At which time step does the ice sheet model receive its input (daily or monthly)? Please state it explicitly.

---

## Author Response (AR1)

The authors would like to thank the editor Thomas Mölg, the two anonymous reviewers, and Irina Rogozhina and Raymond Sellevold for their critical reviews that have have lead to significant improvements of this manuscript.

In addition to a few stylistic changes to the text, we have made the following slightly more substantial changes based on the reviews:

1. We have added a paragraph in the introduction that motivates the simplified modeling approach and puts some of the compromises we make in perspective with the real glacial cycle.
2. We have significantly revised and extended the discussion section to put our results in perspective with previous modeling studies, and to dig a bit deeper into reasons for the increased temperature signal that hinders ice from forming in certain key regions.
3. All figures have been re-plotted on a stereographic projection and with a better suited color scales, and we have added a figure (new Fig. 1) of 500 hPa eddy streamfunction, zonal wind, and total cloudiness to illustrate how the atmospheric dynamics and physics respond to the resolution change.
4. The table has been extended to also include information about the computational demand at the different resolutions (explored in the discussion section).
5. A supplementary document has been added to show fields supporting our conclusions, but that not imperative to follow the main storyline.

**Editor:**
**I would like to thank the two anonymous reviewers and Irina Rogozhina and Raymond Sellevold for their comments and critical reading. My following remarks are thought to help in putting together your final responses.**

**Based on all the comments, there will have to be some expansions to the manuscript, however without having to give up on the general character of this study of being presented in a rather concise format (which I support). I suggest to pay attention to two points in particular.**

**1) A bit more analyses of the results (as I assumed in my access review). – I would in any case suggest to tackle the lapse rate point by sensitivity analyses, and elaborate a bit more on the atmospheric model differences (causes).**

Response:
A new figure has been added to the main text (new Fig. 1), illustrating how the atmospheric "dynamics" (500 hPa eddy streamfunction and zonal wind) and "physics" (total cloudiness) respond to the horizontal resolution. These fields are discussed in relation to previous studies of resolution dependence in Earth-system models. We have also added a supporting document that presents a number of additional figures that help the interpretation (but not essential to

follow the narrative of the main text) of the climate response at the different horizontal resolutions.

We do not fully agree with the criticism regarding the choice of atmospheric lapse rate. First of all, the value 4.6 K/km is misleading as it appears to be based on a subset of the dataset presented in the cited paper (Fausto et. al 2009). When the authors include all data points they get a July lapse rate of 5.5 K/km over the Greenland ice sheet. Secondly, it is not clear that modern day Greenland is a good analog for the continental ice sheets in the last glacial cycle. The glacial climate was generally colder and drier than present, which should shift the temperature lapse rate in the direction of the dry adiabat (9.8 K/km). Case in point, Loomis et al. 2017 (http://advances.sciencemag.org/content/3/1/e1600815) argues that the atmospheric lapse rate in equatorial Africa shifted from 5.8 K/km in the modern climate, to 6.7 K/km at the LGM. Note that this example is from the tropics where both the relative and absolute humidity are expected to be higher than over mid- to high-latitude ice sheets. Thirdly, there is no consensus in the literature that one lapse rate is a better choice than another, as models typically have a range of different tuning parameters that can be used to offset the influence of the atmospheric lapse rate. For example, Stone et al (2010) reproduced the modern day Greenland geometry using lapse rates as different as 4 and 8 C/km, by instead adjusting the largely unconstrained PPD factors accordingly. Given this uncertainty and the fact that we don't actually know what the conditions were like over the Laurentide and Eurasian ice sheets, using the standard (global average) atmospheric lapse rate seems like a reasonable compromise that is in line with the simplified theme of this study. For that reason, running sensitivity experiments with different choices of lapse rates is not deemed necessary as the same argument can be used for all other unconstrained tunable parameters in both the atmosphere and ice sheet models, which would require a huge amount of work to explore properly, and no doubt obscure the main storyline. Having said that, we have added a paragraph in the discussion section on how the choice of lapse rate may influence the climate forcing.

**2) More justification on the study design, see in particular the comment by Irina Rogozhina (and partly RC2). – I would like to add to this issue that the paper would also profit from clarifying the term "coupled" in the given context. Your study is clearly "oneway" (or "standalone", as you say in the paper) with regard to the ice sheet model. Most probably, for the general reader "coupled" would imply "interactive". Therefore, (a) be cautious with using "coupled" here (e.g., not in the title, see RC2 as well) and (b) provide a short argument early in the paper why standalone simulations make sense and/or are still state-of-the-art in face of the comments by Irina Rogozhina. RC2 also suggests in this regard you could argue with the objective of your study.**

Response:
This is a good point. We have changed the title accordingly.

A paragraph has also been added at the end of the introduction to motivate the experimental design, and to highlight some of the simplifications we make in relation to the last glacial cycle. In short, because of the challenges of running coupled climate-ice-sheet model experiment over glacial cycles (identified in the introduction), we scale down on the realism and resort to a simplified experiment design. Although this abstraction makes the study somewhat academic (and unfortunately somewhat esoteric), it is actually beneficial in order to cleanly isolating the influence of the atmospheric resolution on the subsequent ice sheet model experiment. We feel that the addition of this paragraph is a great improvement of the study, as the objectives and general approach are motivated at an early stage.

**Small things to consider:**

**(i) Please explain/argue briefly why only 12 years are simulated with the AGCM. For the general reader, this might be unclear since you talk in the introduction about ice sheet response time scales of hundreds to thousands of years.**

Response:
As stated above, a paragraph has been added to the introduction to motivate the experimental design and put this in perspective with the real glacial cycle. A number of reminders of the study objective have also been added to the text to make it clear that the study should be viewed in a somewhat abstract light.

**(ii) At which time step does the ice sheet model receive its input (daily or monthly)? Please state it explicitly**

Response:
We have added a clarifying statement that the ice sheet model receives monthly climatological forcing data.

**Reviewer 1:**

**This short manuscript presents an assessment of the influence of atmospheric model resolution in coupled climate-ice-sheet simulations. It shows that the atmospheric resolution matters enormously for an accurate simulation of the major LGM ice sheets. The manuscript is concise, clearly written, easily readable and presents an important, albeit un-surprising, result. This study can prove to be an easy to read and quick to fall back on article when introducing new and old glaciologists fresh into these kinds of simulations.**

**My main concern with the manuscript is whether the differences between the different atmospheric resolutions is of dynamical/physical nature, or just a matter of resolution and the model topography. The authors argue that the main cause of accurate/inaccurate simulation of the ice sheets is an inaccurate temperature field. Differences in precipitation are rather small. I think that temperature is a very straightforward parameter to model correctly: it largely depends on elevation, which directly depends on the model resolution. Therefore, this conclusion could have been found without doing any of the model simulations presented.**

Response:
Simulating the temperature is actually harder than one might think as it depends on a range of physical and dynamical processes, such as the surface energy balance, and heat-flux convergence (essentially temperature advection) from the large scale atmospheric flow. Even comparatively small changes in these fields can a have large influence on both the local and global surface temperature profiles. The revised manuscript includes a new figure showing how fields typically associated with the model dynamics (500 hPa eddy streamfunction and zonal wind) and the model physics (total cloudiness) change with the horizontal resolution (new Fig. 1). We have also added a number of supplementary figures exploring sources for the temperature increase at the lower resolutions.

The supplementary figures show, e.g., how the surface temperature over the LGM ice sheets compare to the T85 case when being extrapolated to the modern topography using two different lapse rate corrections (the 6.5 C/km used here, and a lower value of 4.6 C/km that has been observed over the Greenland ice sheet in modern times). This extrapolated temperature is essentially what the ice-sheet model sees upon initialization. The extrapolated temperature, which effectively eliminates the elevation change between the different resolutions, shows a generally similar pattern as the full temperature field (Fig. 2 in the revised manuscript). The lower resolution cases typically have warmer temperatures over the areas covered by the LGM ice sheets, which we interpret as being related to increased downwelling of longwave radiation at the surface due to a general increase in cloudiness. The atmospheric dynamics appears to play a more secondary role for the surface temperature signal (also shown in the supplementary material).

**How robust are the temperature anomalies shown in Fig. 2? Are these resolution differences also found for other atmospheric models (used for ice-sheet forcing)? Or is it CAM3-specific? The authors should make this finding more convincing by comparing (more) the results to other studies. Currently, in the discussion section, much attention is on the quality of the T21 forcing. I would like to see some more focus, in this section, on the intermodel resolution differences. In order to warrant publication this concern should be addressed and made less qualitative.**

Response:
This comment hits at the very heart of the study. As stated in the introduction and in the discussion section, the apparent degradation of the model climate we find between the T31 and T21 resolutions appears to be largely independent of model physics, and therefore rather general among models. Polvani et al (2004) showed that numerical convergence for baroclinic waves is compromised somewhere between the T42 and T21 resolutions, Magnusdottir and Haynes (1999) narrowed it down further and showed that the limit for a "realistic" representation of planetary wave propagation falls somewhere between T31 and T21, and Lofverstrom et al (2016) showed that even the climatological (time mean) circulation is compromised at the T21 resolution. Also, the apparent similarity in simulated ice sheets between our T21 case and CLIMBER-2 suggests that this particular model may suffer from similar shortcomings as we discuss here.

In the revised manuscript we have extended the discussion section to point out similarities with previous studies (we could only find studies that have looked at resolution dependence in the pre-industrial climate). We have also extended the discussion section to be more quantitative, and added a supplementary document where additional comparisons of the different resolutions have been made.

**Possibly some of the following questions could be addressed in more detail: What do the results of this study say about current studies of coupled atmosphere-icesheet models? Have other studies been conducted with inaccurate climate forcing; is this a big issue or not? Have other studies attacked and/or addressed the simple temperature discrepancies; possibly by additional topography (down)scaling techniques, spectral diffusivity, lapse rate corrections? Does the current glaciological community realize that the atmospheric resolution is as important for the results of these types of simulations? How large is the trade-off between "accuracy" and "speed"?**

Response:
The revised discussion section presents a more general discussion on the trade off between "accuracy" and "speed"; the table that has been extended to compare the computational demand at the different resolutions (numerical operations per simulated year, normalized by the T21 case).

In addition, we have also included a discussion on: (1) the robustness of these results with respect to other models (this appears to be the first study on this particular topic so we compare our results with similar studies of the modern climate), and (2) shortcomings of this particular study, and that more refined methods may help improve some of these results. However, we also raise the important point that the quality of the atmospheric climate forcing is ultimately determined by the model resolution, hence it is doubtful that a more realistic ice sheet would be simulated in, e.g., the T21 case, even if using a more sophisticated coupling between the climate and ice sheet models, as the model climate is heavily compromised at this resolution.

Whether or not previous studies have used inaccurate climate forcing is hard to say. All models are simplifications of reality, so on one level the answer is definitely yes. However, models can be inaccurate in different ways (and to different degrees depending on their complexity) as they often use vastly different physics parameterisations. The point we are trying to make when comparing the apparent similarities between our T21 case and CLIMBER-2 is that this particular model appears to suffer from similar deficiencies (perhaps related to the low resolution) that comprehensive circulation models also show when run at very coarse horizontal grids. Whether or not this is an accurate assessment is hard to say without doing a more thorough comparison of these models.

**A different approach might be to use different atmospheric climatologies at several model resolutions to make the results more robust, but I understand if this might prove to be beyond the work of this study.**

Response:
Extending the study to include atmospheric simulation on different types of grids (Guassian, finite difference, finite volume, spectral element, etc.) would allow us to say something more conclusive about the importance of model resolution in these types of experiments. However, that is far beyond the purpose of this study and would require a huge computational effort --- comparable to, or perhaps even beyond the scale of PMIP --- to carry out. Forcing the ice-sheet model with atmospheric data from the PMIP archive would be possible, but it is not obvious how that would benefit this particular study as the participating institute only used one resolution of their specific model. This is however an excellent topic for a potential follow up study.

**Because I do like the brevity of this manuscript and the concise and to the point explanation of this problem, I do suggest publication after minor revisions. Hopefully the authors can tackle my concerns adequately. I have no additional minor comments, except for a typo on p6, line 6: considerable-> considerably.**

Response:
Done

**Reviewer 2:**

**Review of "A note on the influenced of atmospheric resolution in coupled climateăâĨice-sheet simulations":  The manuscript examines the effect of atmospheric resolution on ice sheet modeling forced with climate model output. The theme is certainly relevant for the emerging research on coupled ice-sheet/climate modeling, both in the context of future climate projections as well as paleo-research, and timely due to launch of international collaborative projects such as ISMIP6. To my knowledge, this topic has not been directly addressed in a systematic way like this before.**

**The method applied consists in forcing an ice sheet model (SICOPOLIS, using the Shallow Ice Approximation), with climate output of different resolutions corresponding to the LGM. The ice sheet model is started from zero thickness. PDDs are applied for the surface mass balance calculation.**

**General comment: I would like to see more analysis of the climate model in addition to see the results of applying it as forcing to the ice sheet model. I'll explain in the following. The outcome of the study, namely the identification of a threshold resolution beyond what the climate simulation quality is compromised, is a very interesting result. For this reason, I would urge the authors to expand on the reasons (that is, physical processes lacking, misrepresented, and/or well-captured in the model at each resolution) for this threshold. In other words, what makes the low-resolution model unable to capture the essence of the LGM climate? Introduction: The topic is very well introduced/motivated and the literature review is a great piece of work.**

Response:
A thorough analysis of why the model climate breaks down somewhere between the T31 and T21 resolutions is not an easy task with a model of this complexity. The model used here employs the same parameterizations (solves the same equations) regardless of horizontal resolution, hence the abrupt changes that occur between the T31 and T21 grids appear to show a fundamental limitation of these types of models, rather than changes in the model equations. Similar results have been observed in experiments with both primitive equation models (dry dynamical core without any physics) and comprehensive circulation models (sophisticated physics), suggesting that the numerical breakdown is somehow related to the model dynamics. Since the dynamic equations are scale independent, we speculate that it instead has something to do with the precision of the spherical harmonic transform functions that these dynamical cores rely on. Although we are unable to point to one specific cause of this phenomenon, it has been suggested that the increased diffusion required to ensure numerical stability on coarse horizontal grids is the main culprit. We can neither confirm nor deny this hypothesis from these experiments, but since a similar resolution limit has been shown to influence the development of individual cyclones (Polvani et al. 2004), the way planetary waves propagate (Magnusdottir and Haynes 1999), and also the climatological circulation (Lofverstrom et al. 2016), the model dynamics appears to be compromised on all temporal and spatial scales, and it is therefore reasonable to assume that diffusion is to blame.

We have extended the analysis in the revised manuscript to better support our conclusions. In addition to improve the figures shown in the original submission, we have also added a new figure (Fig. 1) illustrating how the atmospheric "dynamics" (500 hPa stationary waves and zonal jet stream) and "physics" (cloud cover) respond to the horizontal resolution. The figure is discussed and put in perspective with previous modeling studies investigating the effects of resolution in climate-modeling experiments. We have also added a number of supplementary figures to illustrate how other fields respond to the grid resolution.

**Comment on methodology: The method relies in strong assumptions and approximations. The simulation target is to reproduce the reconstructed LGM ice sheets (by Peltier et al.) by building them from zero thickness under a steady-state LGM forcing. In reality, there was a history of building up these ice sheets, so they are not the effect of a constant LGM climate. However, the method seems an efficient reasonable approach for the objective of the manuscript, and seems to work in the identification of a threshold for "minimum required resolution".**

Response:
That is correct. The simplified modeling approach takes a few steps away from reality as we ignore the glacial history, variations in insolation and greenhouse gas concentrations and vegetation, as well as changes in the atmosphere and ocean circulation over the ~100 kyrs from inception to the LGM. However, since the objective of the study is to illustrate that both the ice sheet expansion and the quality of the atmospheric forcing data are strongly controlled by the atmospheric model resolution, it is actually beneficial to resort to a simplified experiment design as many of the challenges with coupled climate--ice-sheet models can be circumnavigated. The drawback is of course that the study becomes somewhat academic (perhaps even esoteric), but that is deemed a necessary compromise to illustrate this phenomenon. A paragraph motivating the study objective and experiment design has been added to the introduction.

**Other comments: It is difficult to follow the precipitation discussion due to the choice for the color bar. Polar latitudes have low precipitation rates, please use a suitable color bar, albeit the loss of resolution for the tropical area.**

Response:
All figures have been revised to have stereographic projections and better suited color scales to highlight the main points of the study.

**I would remove the words "a note" from the title after expanding the manuscript with further climate model analysis. Also, the study does not include "fully" coupled climate Tice-sheet simulations in the sense ˘ that the climate model is not influenced by the ice sheet model in any way. The authors probably chose the wording "in coupled (. . .) simulations" in the context of motivation, but the title can be misleading about the**

**content of the actual study. I would replace the title for e.g. "On the influence of atmospheric resolution on climate-model-forced ice sheet simulations"**

Response:
That is a very good suggestion. We have changed the title accordingly.

**Irina Rogozhina:**

**I have major concerns about the experimental design of this study and how the authors' choices affect the main conclusions of their manuscript. One of such choices is the lapse rate correction. A temperature lapse rate correction is used to derive the temperature forcing at the beginning of ice sheet simulations, which are initiated from the present-day ice-free topography (excluding the Greenland Ice Sheet), and to adjust the temperature forcing to the growth of ice masses throughout the simulations. I question both (1) the choice of the lapse rate correction and (2) the assumption of the initial ice-free conditions, which are undermining the very core of this study.**

**Problem (1): While the authors discuss potential impacts of PDD parameters, they do not question their choice of the temperature lapse rate (6.5C/km). Fausto et al. (2009) measured lapse rates as low as 4.7 and 4.6C/km during the June and July months, respectively (the months of the strongest ablation) on the Greenland ice sheet. This is nearly 2C/km below the value that the authors use throughout the year. The temperature lapse rate is used by the authors to correct the model-based air (or surface) temperatures for the difference between their LGM topography in the atmospheric simulations and ice-free present-day topography assumed at the beginning of their ice sheet simulations. This initial temperature forcing is crucial to the development of an ice sheet: If it is excessively above 0C during the summer period, an ice sheet will not build. A difference of ~2C/km would reduce the corrections of the near-surface temperatures over the areas covered by the former British-Irish and Fennoscandian ice sheets by 3-4C during the months that matter most for the ice sheet surface mass balance, but would not impact Arctic Siberia. Quoting from the study of Löfverström and Liakka, page 1, line 9: "Sensitivity experiments using different surface mass balance parameterizations improve the simulations of the Eurasian ice sheet in the T42 case, but the compromise is a substantial buildup in Siberia". This compromise does not have to be made: The choice of a lower lapse rate correction will trigger a buildup of ice masses over the British Isles and Fennoscandia when using the T42 & T31 climate datasets, reasonable choices of PDD parameters and a higher ice sheet model resolution (see below) but will still keep Arctic Siberia ice free.**

Response:
Although we agree that an annually fixed lapse rate is artificial, it is not at all obvious that 4.6 C/km would be a better motivated choice than using the standard atmospheric lapse rate of 6.5 C/km. First or all, the comment is misleading as the value 4.6 C/km is based on a subset of the data presented in Fausto et al. (2009). When accounting for all data points the authors yield a lapse rate of 5.4-5.5 C/km for the summer months (their Table 3). Second, it is reasonable to assume that the glacial climate was drier than present (response to cooler temperatures via the Clausius-Clapeyron equation), which shifts the lapse rate towards the dry adiabat (9.8 C/km). Note that both proxy data and modeling studies show that the LGM climate was significantly cooler than pre-industrial (the fully coupled LGM simulation in PMIP2 bracket a cooling of -3.7 to -5.8 degC relative to pre-industrial; Braconnot et al. 2007). For example Loomis et al. 2017 (http://advances.sciencemag.org/content/3/1/e1600815) showed evidence that the LGM lapse

rate in tropical Africa was around 6.7 C/km, which is to be compared with the modern value of around 5.8 C/km. Also, even though modeling studies have shown that the lapse-rate correction may be used as a tuning parameter to get "reasonable" ice geometries, there is no consensus in the literature that one value would be a better choice than another. For example Stone et al (2010) simulated a realistic modern day Greenland geometry using lapse rates as different as 4 and 8 C/km, by instead treating the PPD factors as tuning knobs. Given this uncertainty, we maintain that this comment is largely unsubstantiated and that the standard atmospheric lapse rate is an appropriate choice for the objective of this study.

Having said that, there is no doubt that the atmospheric lapse rate may be used to improve the ice sheet expansion in different regions. However, the point of the study is not to be as realistic as possible, but rather to show that the resolution of the atmospheric model (all else being equal) is a potential source of uncertainty in experiments of this type (coupled or semi-coupled climate-ice-sheet simulations). Also, because of the challenges running a comprehensive model over glacial timescales (raised in the introduction), we instead resort to a highly idealized setup to demonstrate this point. It is regrettable that this makes the study somewhat esoteric, but that is unfortunately the nature of the problem.

Lastly, we have extended the discussion section to include a paragraph on ways to improve coupled climate--ice-sheet model experiments. We have also added a (supplementary) figure of the JJA surface temperature extrapolated down to the modern topography (effectively the surface temperature the ice sheet model sees upon initialization) in the different cases using the 6.5 and 4.6 C/km lapse rates. Positive JJA temperatures are found in Eurasia at the T31 and T21 resolutions even when using the lower atmospheric lapse rate.

**Problem (2): A more important question is whether such corrections should be applied at all. While talking about "the influence of atmospheric model resolution in coupled climate-ice sheet simulations" (quoted from the title of the paper), one would rather think about whether the use of lower atmospheric model resolution contaminates the climate state in such a way that it becomes inconsistent with the modeled/prescribed ice sheet geometries (included as a topographic boundary condition in a climate model). The question is not whether this climate forcing can build ice sheets if it's heavily modified using lapse rate corrections but whether it can maintain reasonable ice sheet geometries when unmodified atmospheric model outputs are used.**

Response:
We agree that it would be advantageous not having to use lapse-rate corrections at all, however this is not a viable option as it would require the use of a synchronously coupled climate-ice-sheet model, where all components are discretized on the same grid (projection and resolution), using a realistic SMB calculation based on the local energy balance. Any deviation from this setup will require interpolation and/or lapse rate corrections to be made. However, as explicitly stated in the first part of the introduction, such a simulation is far beyond the current

computational limit of all Earth-system models, and this is not expected to change in any foreseeable future.

This second part of this comment seems to be missing the point of the study. Whether or not the LGM ice sheets can be maintained when using the climate simulated at different horizontal resolutions is not what we are examining here. This is however an excellent topic for a follow-up study.

**While Problem (1) can be easily tackled by performing additional sensitivity experiments, Problem (2) is more challenging to resolve. The authors could test T42, T31 and T21 datasets using ice sheet modeling results from the T85 dataset "reproduced to a high accuracy" (quoted from Löfverström and Liakka, page 1, line 8) to address the question, whether the degradation of the atmospheric model resolution results in ice sheet collapses consistent with their current conclusions. However, their modeled ice sheets in the T85 simulation are 1.5 to 2 km too thick relative to the existing reconstructions of the LGM ice sheet geometries. Even without additional lapse rate corrections (introduced to reconcile the difference between the ice sheets prescribed in the atmospheric simulations and derived from the T85 simulation), removing such thick ice sheets would be a difficult task for the T42, T31 and even T21 data sets. At this point a question arises: Why are the modeled ice sheets so unrealistically thick? I envision several potential causes of such unrealistic model performance: (i) The spin-up of the ice sheet model: Running an ice sheet model to an equilibrium with the LGM climate over 150 thousand years is not in line with the existing evidence. Most of the former ice sheets were short-lived (tens of thousands of years from buildup to decay) (ii) Shallow ice approximation (excluding ice stream dynamics) in combination with excessively low resolution of the ice sheet model (80 km) fails to approximate the rapid ice flow and routing of ice masses towards the ocean. (iii) The isostatic adjustment scheme may cause an exaggerated bending of the bedrock surface under the weight of growing ice sheets (I have not seen the Local Lithosphere and Relaxing Asthenosphere method being used in years).**

Response:
Not exactly. The figure (Fig. 4 in the revised manuscript) shows ice thickness, not topography above sea level. This means that approximately 30% of the quantity shown is depressed below the modern bedrock elevation. With a maximum North American ice thickness of about 5-6 km, this translates to an actual topographic height of 3.5-4 km, which is in broad agreement with modern reconstructions of the LGM Laurentide ice sheet. For example, the highest point over the Laurentide ice-sheet topography in ICE-6G is around 3900 m.

The suggested improvements are orthogonal to the main objective of the study. We are not trying to make a realistic simulation of the ice evolution over the last glacial cycle, but rather to show a previously unexplored source of bias (importance of atmosphere model resolution) that we think that the research community interested in climate--ice-sheet model experiments should be aware of. That our methodology deviates from reality is no secret and we state this explicitly

in the paper, however whether or not these simplifications lead to false conclusions is something future research will show.

**The overall quality of the study could improve if the authors address problems (1) and (2). It can also benefit from the use of SICOPOLIS v3.3 that includes options for higher resolution, more realistic treatment of ice streams and glacial isostatic adjustment. Finally I strongly suggest that the authors improve their figures. The adopted projection strongly distorts the Arctic region, which is the main focus of the present study.**

Response:
For the scope of the study we do not think that a newer version of SICOPOLIS would be advantageous. The projection and color scales were arguably somewhat lacking in the original submission and we have re-plotted all figures on stereographic projection with a refined color scale.

**Raymond Sellevold:**
**You mention in the introduction how horizontal diffusion does not only influence horizontal motions, but may also impact vertical transport and convection. I wonder how important resolving convection adequately is for building/removing an ice sheet. It would be informative if you more specifically related resolution dependent dynamics/physics with possible shortcomings in building/removing the ice sheets.**

Response:
The precipitation in the midlatitude storm tracks is predominantly large scale (stratiform), so a weakening/breakdown of the convection (used loosely here as these types of clouds are highly parameterised in climate models) is probably not a major influence on the surface mass balance by itself. However, both atmospheric convection and large-scale vertical transport are important components of cyclogenesis, so the large-scale precipitation field may be altered via more indirect processes related to vertical motions. Also, the fact that the quality of virtually all atmospheric fields is compromised at sufficiently coarse horizontal grids suggests that there may be a fundamental problem with low-resolution models that transcends both model tuning and complexity.

We have added figures comparing the large scale circulation to illustrate how the different resolutions compare to one another. Fields related to both model dynamics and physics are shown and discussed in relation to previous studies.

**In your method section, you describe that present day non-glaciated areas are prescribed with modern day vegetation cover. Do you think this assumption is valid? Do you think it may have a large influence on the simulations?**

Response:
The pre-industrial vegetation is almost certainly impacting the ice extension, even though the influence should be largely comparable at all resolutions and therefore not of first order importance for our main conclusions. No reliable reconstruction of the LGM biome exists, so the modern vegetation cover is actually used in the official PMIP boundary conditions (https://pmip4.lsce.ipsl.fr/doku.php/exp_design:lgm), unless the model can simulate vegetation changes interactively (not an option in CAM3).

Note that the AGCM simulations have prescribed ice sheets in the areas indicated by the black contours in the figures, hence the surface albedo is high in those areas regardless of the model resolution. Differences in the simulated ice sheets (e.g. in western Eurasia and to some extent in the interior of North America) is therefore not to first order a response to the (modern) vegetation cover.

We have added a clarifying statement about the use of modern vegetation in the text, and also cited the paper outlining the official PMIP4 LGM simulation design.

**You present the smoothed topography as a reason why some areas are warmer when the resolution is lower. After that, areas that are colder with respect to the T85 case are pointed out with no explanation why they may be colder.**

Response:
There can be any number of explanation for local positive and negative anomalies in the difference fields. In addition to differences in the simulation quality at the different resolutions (note that the objective of the study is to illustrate that there is a general degradation of the simulation quality on coarser horizontal grids), the short (10 year) climatologies are certainly one possible explanation for these anomalies. Also, comparing gridded data on different resolutions can naturally give rise to small scale "inconsistencies" like that. Note that there is only one value of temperature in each grid cell, so even if the 4x4 cluster on the T85 grid that covers one cell on the T21 grid has the same average temperature as the T21 cell, grid cells one side of the cluster might be cooler than the average, while the other side might be warmer. Subtracting the two would then yield local warm and cold anomalies even if the area means are the same. Examining this further is however beyond the scope of the study as we try to focus more on the large scale structure, rather than local scale variations in the difference fields.

**The tropical and midlatitude precipitation fields are well covered in the text, but there is not much mentioning of precipitation over the ice sheets. It is also hard to see the difference of this field between different resolutions, since the colors starts at 200 mm/yr. A possible solution would be to make a non-linear color scale to better resolve the low-precipitation areas (such as the ice sheets).**

Response:
That is a fair point. We have changed the projection and the color scale to better illustrate the changes over the ice sheets. We have also made appropriate changes to the text where we describe the precipitation field.

**"The ice sheets forming under the high resolution atmospheric climatology (T85; panel 3a) are in close resemblance with the target extent (indicated by solid contours; Kleman et al., 2013). There is essentially only too much ice extending along the Siberian Arctic coast." Why is there too much ice extending there? Because it is very cold, because the precipitation is very high, or maybe a combination of both?**

Response:
It is probably a combination of both, even though the temperature field is the most important contributor. Explaining the cause is however outside the scope of the study and therefore not examined further.

**"However, the T21 resolution only has "functional support", which means that boundary conditions are provided but the model climate has not been tuned to the same standard as the other resolutions (the resolution dependent tuning parameters are broadly the same as in the T42 case)." How is the 1850-present climate in this low resolution as it is not tuned? Was there any attempt made to tune it? If no, why not?**

Response:
Well, the climate is tuned but not to the same rigour as the other resolutions. Also the modern (pre-industrial) climate simulated on the T21 grid is quite different from observations (not shown).

No attempt was made to tune the T21 climate. Even though CAM3 is a comparatively simple model with modern day standards (e.g. CAM6, the newest addition to the NCAR model family, is much more complex), it is still a comprehensive AGCM with twelve independent tuning knobs. Re-tuning such a model is a herculean task that is extremely difficult and therefore typically carried out by a team of experts.

**On the influence of atmospheric resolution on climate-model-forced ice sheet simulations**

\Author[1]{Marcus}{Lofverstrom}
\Author[2]{Johan}{Liakka}

\affil[1]{National Center for Atmospheric Research,  3090 Center Green Dr., 80301, Boulder, CO, USA}
\affil[2]{Nansen Environmental and Remote Sensing Center, Bjerknes Centre for Climate Research}

\begin{abstract}
Coupled climate--ice-sheet simulations have been growing in popularity in recent years. Experiments of this type are however challenging as ice sheets evolve over multi-millennial time scales, which is beyond the practical integration limit for most Earth-system models. A common method to increase model throughput is to trade resolution for computational efficiency (compromises accuracy for speed). Here, we analyze how the resolution of an atmospheric general circulation model (AGCM) influences the simulation quality in a standalone ice-sheet model. Four identical AGCM simulations of the Last Glacial Maximum (LGM) were run at different horizontal resolutions: T85 (1.4$^{\circ}$), T42 (2.8$^{\circ}$), T31 (3.8$^{\circ}$), and T21 (5.6$^{\circ}$). These simulations were subsequently used as forcing of an ice-sheet model. While the T85 climate forcing reproduces the LGM ice sheets to a high accuracy, the intermediate resolution cases (T42 and T31) fail to build the Eurasian Ice Sheet. The T21 case fails in both Eurasia and North America. Sensitivity experiments using different surface mass balance parameterizations improve the simulations of the Eurasian ice-sheet in the T42 case, but the compromise is a substantial ice buildup in Siberia. The T31 and T21 cases are not improving in the same way in Eurasia, though the latter simulates the continent-wide Laurentide Ice Sheet in North America. The difficulty to reproduce the LGM ice sheets in the T21 case is in broad agreement with previous studies using low-resolution atmospheric models, and is caused by a substantial deterioration of the atmospheric climate between the T31 and T21 resolutions. It is speculated that this deficiency may demonstrate a fundamental problem using low-resolution atmospheric models in these types of experiments.
\end{abstract}

%\copyrightstatement{TEXT}

\introduction \label{sec:introduction} %% \introduction[modified heading if necessary]
%

Experiments with coupled climate--ice-sheet models have become increasingly popular in recent years, much thanks to coordinated international modeling initiatives such as the ``Ice Sheet Model Intercomparison Project'' (ISMIP6) \citep[][]{NowickiEA2016_ISMIP6} and the ``Pliocene Ice Sheet Modelling Intercomparison Project'' (PLISMIP) \citep[][]{DolanEA2012plismip}. These types of experiments are challenging as ice sheets have a high thermal inertia that makes their response time considerably greater than all other components of the climate system---the time scale depends on the application but it typically ranges from $10^3$ to $10^5$ years. Simulations of this length are beyond the practical integration limit of most Earth-system models, and different techniques to bypassing this problem have therefore been devised. Some of the more popular approaches for simulating ice sheets over glacial time scales include:
%
\begin{itemize}
\item[(i)] Force a standalone ice-sheet model with a transient climate record obtained by interpolating between the climate extremes over the period of interest (often simulations of the pre-industrial and the Last Glacial Maximum; PI and LGM, respectively). The interpolation weights are typically derived from oxygen isotope ratios in Greenland and Antarctic ice cores \citep[e.g.][]{CharbitEA2007,FykeEA2014gmd}.
%
\item[(ii)] Use an asynchronous coupling between an ice-sheet model and a general circulation model (GCM). The ice-sheet model, which is computationally cheaper than the GCM, is then run multiple years between each update of the model climate \citep[e.g.][]{LiakkaEA2011,Liakka2012,HerringtonPoulsen2012jclim,LofverstromEA2015JClim}.
%
\item[(iii)] Utilize a computationally efficient intermediate complexity model (EMIC) that can be run transiently over glacial time scales \citep[e.g.][]{RoeLindzen2001,CalovEA2005part1,BonelliEA2009,GanopolskiEA2010CP,BeghinEA2014}.
%
\end{itemize}
%
Although no attempt is made here to assess how these different approaches compare to one another, we conclude that they all rely on a number of assumptions and simplifications that potentially can influence the results. For example:
%
(i) assumes that the glacial climate changes as a linear combination of the PI and LGM states, which is at odds with both modeling and proxy-data evidence of highly nonlinear circulation changes over the last glacial period \citep[e.g.,][]{Jackson2000,ZhangEA2014,LoraEA2016grl,PausataLofverstrom2015grl,LofverstromEA2016jas,LofverstromEA2014,LofverstromLora2017grl};
(ii) accelerating the ice-sheet component introduces abrupt changes in the GCM boundary conditions, which may force the model climate into an unphysical state at the beginning of each (GCM) run segment;

(iii) simplified models often rely on statistical dynamics/physics where almost all interactions are prescribed or represented by first-order linear assumptions.

In addition, one issue that has received little attention in the literature is what role the atmospheric grid resolution---the horizontal mesh on which the model equations are discretized---plays in coupled climate--ice-sheet experiments. Simplified circulation models often utilize a coarse horizontal grid for computational efficiency. For example, the atmospheric component of CLIMBER-2 has a horizontal resolution of approximately $10^\circ\times51^\circ$ \citep[][]{PetoukhovEA2000ClimDyn}, LOVECLIM runs on a $5.6^\circ\times5.6^\circ$ resolution grid \citep[][]{GoosseEA2010gmd}, and FAMOUS on a $5^\circ\times7.5^\circ$ grid \citep[][]{SmithEA2008gmd}. These are to be compared with the $1^\circ\times1^\circ$ operational resolution of many modern GCMs \citep[e.g.][]{FlatoEA2013_AR5_Ch9}.

[revised manuscript text omitted]
 \citep[][]{Cuemas_Cordon_2011,Hourdin_EA_2012, Demory_EA_2014_climdyn}, and thus appears  to be a fairly robust feature, largely independent of grid type and physics parameterizations.

The T21 case shows a fairly different response with a considerable warming over most of the world's topography (Fig.\,\ref{fig:ts}g), including the ice sheets. This is likely a response to the lower mean-height of the resolved topography (smoothing from the interpolation process), but also from a general degradation of the model climate and a strongly enhanced downwelling of longwave radiation related to the increased cloudiness  (Fig.\,\ref{fig:psi_cldtot}; see also further discussion in Section\,\ref{sec:Discussion}). The midlatitude precipitation field is also considerably altered with respect to the T85 case, with substantially lower precipitation in the eastern parts of the midlatitude storm tracks and thus over the southwestern parts of the ice sheets (Fig.\,\ref{fig:prect}). Note that this is presumably a response to the model's inability to resolve planetary waves (and hence individual cyclones) at coarse horizontal resolutions \citep[Fig.\,\ref{fig:psi_cldtot}; see also][]{PolvaniEA2004,MagnusdottirHaynes1999,Cuemas_Cordon_2011,Hourdin_EA_2012,LofverstromEA2016jas}.

%%%%%%%%%%%%%%%%%%%%%%%%%%%%%%%%%%%%%%%%%%%

\section{Ice sheet model results} \label{sec:Results}

The left column in Fig.\,\ref{fig:ice} shows the equilibrium ice-sheet extent when using the default SMB parameterization in SICOPOLIS (SICOdef). The ice sheets forming under the high

resolution atmospheric climatology (T85; panel \ref{fig:ice}a) are in close resemblance with the target extent \citep[indicated by solid contours; ][]{KlemanEA2013}, with only slightly too much ice extending in western Canada and along the Siberian Arctic coast.

The ice sheets forced by the intermediate resolution climatologies (T42 and T31; panels \ref{fig:ice}b,c) both adequately reproduce the North American ice sheet, while they fail to build the Eurasian counterpart in agreement with the reconstruction. This one-sided mismatch can be understood from the atmospheric climatologies described in Section\,\ref{sec:climatologies}. The warm summer temperature over the southwestern parts of the Eurasian Ice Sheet (Figs.\,\ref{fig:ts}e,f) is the main reason for why ice is not forming in this region. Note that although there is a relatively small reduction of precipitation with respect to the T85 case (the interior of Scandinavia is actually showing larger values than the T85 case), the warm surface temperatures are by far the most pronounced feature over the Eurasian Ice Sheet (cf. Figs\,\ref{fig:ts} and \ref{fig:prect}; see discussion in Section\,\ref{sec:Discussion}). These results are in broad agreement with \citet{AbeOuchiEA2013nature}, who showed that the Eurasian Ice Sheet is more sensitive to temperature changes than the North American counterpart. The relatively small temperature change over the Eurasian Ice Sheet is thus sufficiently strong to influence the ice sheet response there. The warming signal in northwestern North America is located in a relatively cold region with a short ablation season, and therefore has a smaller influence on the local ice sheet evolution.

The T21 case, on the other hand, struggles to reproduce the LGM ice sheets in both continents. Although ice forms in North America, it fails to build the continent-wide Laurentide Ice Sheet and instead forms two separate ice sheets---a smaller eastern and a larger western dome--- separated by a wide gap in the region around Hudson Bay. This response bears some structural similarity to the low-resolution model results shown in \citet{BeghinEA2014} and \citet{CharbitEA2013}, and also the pre-LGM ice sheets in \citet{CalovEA2005part1}, and \citet{BonelliEA2009}. Similar to the T42 and T31 cases, the T21 climate forcing is too warm (and presumably too dry) over the southwestern parts of the Eurasian Ice Sheet area to reproduce the LGM ice sheet reconstruction.

The sensitivity experiments with different SMB parameterizations in SICOPOLIS are presented in Fig.\,\ref{fig:ice}e-l. The middle row (panels \ref{fig:ice}e,f,g,h) uses the FST09 ablation model, and the bottom row (panels \ref{fig:ice}i,j,k,l) the ablation model  described in TP02. Both these alternative SMB parameterizations help improve the Eurasian ice extent in the T42 case (Figs.\,\ref{fig:ice}f,j), though at the price of a fairly substantial ice buildup in northern Siberia and Beringia (particularly pronounced in Fig.\,\ref{fig:ice}f), which are areas that were largely ice free at the LGM \citep[][]{SvendsenEA2004,KlemanEA2013, LofverstromLiakka2016grl}. A broadly similar buildup in these regions is also seen in the T85 case when using these SMB parameterizations.

The alternative SMB parameterizations are not improving the ice-sheet simulations in Eurasia when using the lower resolution climatologies (T31 and T21; Fig.\,\ref{fig:ice}g,h,k,l), but they help the formation of a continent-wide Laurentide Ice Sheet in the T21 case (Fig.\,\ref{fig:ice}k,l).

%%%%%%%%%%%%%%%%%%%%%%%%%%%%%%%%%%%%%%%%%%%%%%%%%

\conclusions[Discussion and conclusions] \label{sec:Discussion}

The results presented in this paper attempt to illustrate, albeit in a highly qualitative way, the influence of atmospheric model resolution on climate-forced ice-sheet-model simulations. By adopting a simplified modeling approach we can effectively isolate the influence of the atmospheric model, and by prescribing LGM boundary conditions (sea-surface conditions and continental ice sheets) the ice formation is primed to occur in the ``correct'' areas in the subsequent ice-sheet model experiments. This methodology appears to work well when using the high resolution atmospheric climatology \citep[T85; see also][]{LiakkaEA2016cp}, but is less successful when using the climatologies from the lower resolution simulations (T42, T31, and T21; Fig.\,\ref{fig:ice}). There are primarily two explanations for this: (i) lapse-rate effects due to differences in resolved topography; and (ii) changes in the simulated climate that are conducive for warm surface temperatures over the LGM ice sheets. We discuss these processes in the next two paragraphs:

(i) Moving to a coarser horizontal resolution typically results in a lapse-rate induced surface warming, as the resolved topography is both lower and smoother as a result of the increased grid spacing. In this study we employed the modern global-average lapse rate of 6.5\,$^\circ$C\,km$^{-1}$ for vertical interpolation/extrapolation. This is about 1 to 2\,$^\circ$C\,km$^{-1}$ higher than modern observations over the Greenland Ice Sheet in boreal summer \citep[][]{Fausto_EA_2009_JGlac}, but it is motivated by generally drier conditions in glacial climates \citep[Clausius-Clapeyron scaling; LGM simulations typically feature a global cooling of 4 to 6\,$^\circ$C relative to pre-industrial; e.g.][]{BraconnotEA2007} that shift the lapse rate towards higher values; e.g. \citet{Loomis_EA_2017_Science} showed that the tropical atmospheric lapse rate may have increased from around 5.8\,$^\circ$C\,km$^{-1}$ in the modern climate to 6.7\,$^\circ$C\,km$^{-1}$ at the LGM. The elevation difference in the interior of the Laurentide ice sheet is around 200\,m between the T85 and T21 cases (Fig.\,S1). Hence, the lapse-rate effect is accounting for 5-10\% of the local warming signal seen in Fig.\,\ref{fig:ts}. The effect is however more substantial at the ice sheet edges and in Eurasia (accounting for 30 to 50\% of the warming signal), where the elevation difference is larger (Fig.\,S2).

(ii) The majority of the temperature difference in Fig.\,\ref{fig:ts} is induced by changes in the atmospheric circulation. The stationary planetary waves are considerably weaker in the T21 case (Fig.\,\ref{fig:psi_cldtot}), resulting in reduced cold-air advection over the Laurentide Ice

Sheet (Fig.\,S3). The (total) cloudiness is at the same time significantly increased (Fig.\ \ref{fig:psi_cldtot}). While clouds help regulate the amount of downwelling shortwave radiation at the surface, upper level ice-clouds increase the re-emission of longwave radiation back to the surface. Here clouds are found to increase the surface radiative heating effect (SW$_\textrm{net}$ + LW$_\textrm{down}$) by more than 30\,W\,m$^{-2}$ over the ice sheets (Fig.\,S4).

As a result, while the T42 and T31 cases struggle to build ice in Eurasia, the T21 experiment fails to build the continent-wide Laurentide Ice Sheet in North America (when using the default SMB parameterization; SICOdef). Instead it builds two spatially disconnected ice sheets, with a larger dome on the western side of the continent (Fig.\,\ref{fig:ice}d). Several coupled climate–ice-sheet experiments with a low-resolution atmospheric model have shown qualitatively similar results, see e.g.: \citet{CalovEA2005part1, CharbitEA2013, BeghinEA2014}. The common denominator for these studies is that they all used CLIMBER-2 to produce the atmospheric forcing fields. We stress that it is not our intention to single out this particular model, but it appears to suffer from similar deficiencies as our T21 case and may therefore help us understand some of these results. In the aforementioned papers the ice sheet tends to be limited to the western/northwestern side of the North American continent \citep[e.g.][]{CharbitEA2013,BeghinEA2014}, little or no ice is established in western Eurasia \citep[e.g.][]{ CalovEA2005part1,CharbitEA2013,BeghinEA2014}, and attempts to remedy these shortcomings typically result in substantial ice formation in Siberia and Alaska \citep[see][who tested the sensitivity of the same PDD-based SMB parameterizations as were used in this study]{CharbitEA2013}. These results appear to be largely independent of both the choice of ice-sheet model (the above studies used SICOPOLIS and GRISLI), and the complexity of the SMB parameterization \citep[][]{CharbitEA2013,BauerGanpolski2017CP}. Although it is not completely fair to compare CLIMBER-2 to a low resolution version of CAM3 (the complexity and general purpose of these models is extremely different), it is possible that these similarities demonstrate a fundamental problem with low-resolution climate models that transcends model complexity.

One piece of information that is rarely mentioned in the literature is that most Earth-system models are tuned to reproduce the climate of the instrument era ($\sim$1850 to present). These models are of course valuable tools for exploring other time periods as well, but it generally means that inter-model discrepancies tend to increase under more extreme forcing scenarios, e.g. glacial conditions \citep[e.g.][]{BraconnotEA2007}. The results presented here suggest that the model spread may be further exacerbated by differences in horizontal resolution.

The model used here has been tuned and extensively tested at the T85, T42, and T31 grids \citep[e.g.][]{CollinsEA2006CCSM3,YeagerEA2006jclim}. However, the T21 resolution only has ``functional support'', which means that boundary conditions are provided but the model climate has not been tuned to the same standard as the other resolutions (the resolution dependent

tuning parameters are broadly the same as in the T42 case). This is probably at least a partial explanation for the apparent degradation of the model climate at T21, though it is possible that this manifests a more general breakdown of the numerical convergence that has been identified in previous modeling studies \citep[e.g.][]{PolvaniEA2004,MagnusdottirHaynes1999,DongValdes2000}. Some evidence of this is seen in Fig.\,\ref{fig:psi_cldtot}: while the model physics shows a fairly gradual change between the T85 and T21 resolutions (\ref{fig:psi_cldtot}e-h)---including an equatorward migration of the storm track (mid-latitude precipitation) and an increased cloudiness \citep[a similar response to horizontal resolution has been identified in studies of the modern climate;][]{Hack_EA_2006_CCSM3,Guemas_Codron_2011,Hourdin_EA_2012,Demory_EA_2014_climdyn}---fields associated with the model dynamics retain much of their amplitude and general structure at the T31 resolution, but deteriorate significantly when going to T21. What manifests an acceptable simulation quality is subjective and strongly dependent on application. However, since ice sheets are sensitive to feedback loops triggered by deviations from ``expected'' (both mean state and variability) climate conditions, coupled climate--ice-sheet simulations generally require a higher simulation quality than more traditional modeling experiments.

However, resorting to a lower horizontal resolution can increase both the model throughput (simulated years per day), and reduce the simulation cost \citep[number of CPU-hours per simulated year; e.g.][]{YeagerEA2006jclim}. As shown in Table\,\ref{table:cam3_settings}, simulating one model year on the T85 resolution requires around $21\times$ as many operations as one model year on the T31 grid, and $48\times$ as many operations for the same integration length on the T21 grid. This encapsulates the challenges of coupled climate--ice-sheet experiments, as it is common to trade resolution (``accuracy'') for computational efficiency (``speed'') in order to run transient simulations over glacial timescales.

As a final point we note that modern GCM's often utilize highly sophisticated, conservative methods to pass data between the model components. Although it is possible that some of the shortcomings discussed here---e.g. the lack of ice formation in western Eurasia in the T42 and T31 cases, and in east-central North America in the T21 case---may be due to the simplified experiment design and selected parameter values (some hints of this is seen in Fig.\,S2), it is important to stress that the ice evolution is ultimately determined by the atmospheric data, which is strongly controlled by the model resolution. The precise limit where the atmospheric resolution becomes an issue is probably model specific and dependent on application, but a lower bound for most experiments is likely to be somewhere around the nominal T31 resolution, and possibly higher (T42 or even T85) for coupled climate--ice-sheet model simulations.

%%%%%%%%%%%%%%%

\begin{table}[t]
\caption{
%
Resolution specific settings. The top two rows show the nominal horizontal resolution in degrees [$^\circ$] and in number of grid cells [lat$\times$lon], respectively.
The run cost (third row) is normalized with respect to the T21 case and estimates the number of calculations required per model year at the different resolutions. This estimate is based on the grid size and the nominal time step [s] for each resolution.
The horizontal (biharmonic) diffusion coefficient (bottom row) is given in units of $10^{15}$\,m$^4$s$^{-1}$.
}
%
\label{table:cam3_settings}
{\begin{tabular}{rcccc}
\tophline
 & T85 & T42 & T31 & T21 \\
\middlehline
Resolution & 1.4 & 2.8 & 3.8 & 5.6\\
%
Grid cells & 128$\times$256 & 64$\times$128 & 48$\times$96 & 32$\times$64 \\
%
Normalized run cost & 48 & 6 & 2.25 & 1 \\
%
Time step & 600 & 1200 & 1800 & 1800 \\
%
Diffusion & 1 & 10 & 20 & 20 \\
\bottomhline
\end{tabular}}
\belowtable{
}
%}
\end{table}

%%%%%%%%%%%%%%%

\begin{figure}[t]
  \includegraphics[width=1.\textwidth]{PSI_CLDTOT.pdf}
 \caption{
 (Left) Summer (JJA) 500 hPa eddy streamfunction [m$^2$s$^{-1}$] (shading; zonal mean removed) and zonal wind [m\,s$^{-1}$] (contours; 10\,m\,s$^{-1}$ intervals starting at 20\,m\,s$^{-1}$); (right) vertically integrated (total) cloudiness [$\%$].
  %

The 500\,m ice-sheet topography from the LGM reconstruction is indicated by the heavy
contours (interpolated to the different horizontal resolutions).
 }
 \label{fig:psi_cldtot}
\end{figure}

\clearpage

%%%%%%%%%%%%%%%%%%%%%%%%

\begin{figure}[t]
%  \noindent\includegraphics[width=1.\textwidth]{TS.pdf}
        \includegraphics[width=1.\textwidth]{TS.pdf}
 \caption{
  Summer (JJA) surface temperature [$^\circ$C] from the different resolution atmospheric
climatologies. The full fields are shown in the left column (panels a,b,c,d), and the difference
with respect to the T85 case is shown in the right column (panels e,f,g).
 %
  The 500\,m ice thickness from the LGM reconstruction is indicated by the heavy contours
(interpolated to the different horizontal resolutions).
 }
 \label{fig:ts}
\end{figure}

\clearpage

%%%%%%%%%%%%%%%%

\begin{figure}[t]
 \includegraphics[width=1.\textwidth]{CumPrec.pdf}
 \caption{
  Cumulative sum of precipitation (liquid + solid) over the year (total annual amount)
[mm\,year$^{-1}$] from the different resolution atmospheric climatologies. The full fields are
shown in the left column (panels a,b,c,d), and the difference with respect to the T85 case is
shown in the right column (panels e,f,g).
 %
  The 500\,m ice thickness from the LGM reconstruction is indicated by the heavy contours
(interpolated to the different horizontal resolutions).
 }
 \label{fig:prect}
\end{figure}

\clearpage

%%%%%%%%%%%%%%
\begin{figure}[t]
 \includegraphics[width=1.\textwidth]{ice_cyl.pdf}
 \caption{
  Equilibrium ice thickness [m] when using different ablation parameterizations in the surface mass balance scheme: (left) default method in SICOPOLIS; (middle) method by \citet{FaustoEA2009JGlac}; and (right) method by \citet{TarasovPeltier2002}. The atmospheric climatologies from the (a,e,i) T85; (b,f,j) T42; (c,g,k) T31; and (d,h,l) T21 resolution simulations were used, respectively.
  The 500\,m ice thickness from the LGM reconstruction is indicated by the heavy contours (interpolated to the different horizontal resolutions).
 }
 \label{fig:ice}
\end{figure}

---

## Author Response (AR2)

We thank the editor for these comments and suggestions. We have revised the text accordingly, and also changed a small number of other formulations that caught our eyes when reading through the text.

**Thank you for the thorough revision based on the two referee reports and the two short comments. In my opinion you addressed the main points of the comments well, and still maintained the type of a "note" that can stimulate more research in this direction. The revised paper is more convincing in particular due to the new analyses on model physics and dynamics, which clarify the different results for different grid resolutions.**

**Some comments to consider at this point are as follows.**

**P1/L20: Suggest "most" or "almost all" instead of "all" (e.g. the deep ocean is also very slow).**

Answer:
We have changed the sentence accordingly.

**P1/L24: typo; the r is missing in throughput**

Answer:
Thanks for noticing.

**P5/L1: Suggest "the precipitation amount changes"; you mean amount, right? It would help to distinguish from the precipitation fraction (solid/liquid) discussed in the next sentence.**

Answer:
We have added the word "amount" to avoid ambiguity.

**P5/L28: Please find a different expression for "narrower", it is imprecise (e.g., less extensive in meridional/zonal direction?).**

Answer:
We have changed the sentence accordingly.

**Start of Section 5: It is a nice idea to single out the points (i) and (ii) and discuss them in more detail below. Would it also make sense to spend a quick word on precipitation in an item (iii)? If not, I would underline in the first paragraph of 5 that you are focusing on the temperature issue.**

Answer:
We have added a note explaining that the precipitation changes are thought to be secondary, and that we therefore focus on the temperature field.

**Table 1: Could you add as final sentence a note on your diffusion unit, since most readers will think of $m^2/s$ as the standard unit?**

Answer:
The word "biharmonic" means that the nabla-operator is to the fourth power in the diffusion parameterization. We have added a clarifying statement in the table caption.

**Finally, the title: it reflects the paper content better now, but reads a bit like a review-type article heading. Could you narrow down the implications of the title and bring it more toward a case study indication (what your study is)? This would also be in line with comments during the discussion phase, that some settings of your study are still specific and hard to generalize. Some suggestions are:**

**The influence of atmospheric model resolution in CAM3 on climate model-forced ice sheet simulations**

**The influence of atmospheric model resolution in a climate model-forced ice sheet simulation**

**The influence of atmospheric model resolution on climate model-forced ice sheet simulations: Case study with CAM3**

**You are free to suggest something else, but it should read less book chapter-like.**

Answer:
We don't like to specify the AGCM in the title, as we believe that most of these results are general and thus apply to other models as well. We therefore selected the second option of your suggestions.

[revised manuscript text omitted]

In addition, one issue that has received little attention in the literature is what role the atmospheric grid resolution---the horizontal mesh on which the model equations are discretized---plays in coupled climate--ice-sheet experiments. Simplified circulation models often utilize coarse horizontal grids for computational efficiency. For example, the atmospheric component of CLIMBER-2 has a horizontal resolution of approximately $10^\circ\times51^\circ$ \citep[][]{PetoukhovEA2000ClimDyn}, LOVECLIM runs on a $5.6^\circ\times5.6^\circ$ resolution grid \citep[][]{GoosseEA2010gmd}, and FAMOUS on a $5^\circ\times7.5^\circ$ grid \citep[][]{SmithEA2008gmd}. These are to be compared with the nominal $1^\circ\times1^\circ$ resolution of many modern GCMs \citep[e.g.][]{FlatoEA2013_AR5_Ch9}.

M 3/10/18 10:19 AM

[revised manuscript text omitted]
 \citep[Fig.\,\ref{fig:psi_cldtot}; see also][]{PolvaniEA2004,MagnusdottirHaynes1999,Guemas_Codron_2011,Hourdin_EA_2012,Lof verstromEA2016jas}.

%%%%%%%%%%%%%%%%%%%%%%%%%%%%%%%%%%%%%%%%%%%

\section{Ice sheet model results} \label{sec:Results}

The left column in Fig.\,\ref{fig:ice} shows the equilibrium ice-sheet extent when using the default SMB parameterization in SICOPOLIS (SICOdef). The ice sheets forming under the high resolution atmospheric climatology (T85; panel \ref{fig:ice}a) are in close resemblance with the target extent \citep[indicated by solid contours; ][]{KlemanEA2013}, with only slightly too much ice extending in western Canada and along the Siberian Arctic coast.

The ice sheets forced by the intermediate resolution climatologies (T42 and T31; panels \ref{fig:ice}b,c) both adequately reproduce the North American ice sheet, while they fail to build the Eurasian counterpart in agreement with the reconstruction. This one-sided mismatch can be understood from the atmospheric climatologies described in Section\,\ref{sec:climatologies}. The warm summer temperature over the southwestern parts of the Eurasian Ice Sheet (Figs.\,\ref{fig:ts}e,f) is the main reason for why ice is not forming in this region. Note that although there is a relatively small reduction of precipitation with respect to the T85 case (the interior of Scandinavia is actually showing larger values than the T85 case), the warm surface temperatures are by far the most pronounced feature over the Eurasian Ice Sheet (cf. Figs\,\ref{fig:ts} and \ref{fig:prect}; see discussion in Section\,\ref{sec:Discussion}). These results are in broad agreement with \citet{AbeOuchiEA2013nature}, 
[revised manuscript text omitted]
 \citep[CPU-hours per simulated year; e.g.][]{YeagerEA2006jclim}. As shown in Table\,\ref{table:cam3_settings}, simulating one model year on the T85 resolution requires around $21\times$ as many numerical operations as one model year on the T31 grid, and $48\times$ as many operations for the same integration length on the T21 grid. This encapsulates the challenges of coupled climate--ice-sheet experiments, as it is common to trade resolution (``accuracy'') for computational efficiency (``speed'') in order to run transient simulations over glacial timescales.

Lastly, it is possible that some of the shortcomings discussed here---e.g. the lack of ice forming in western Eurasia in the T42 and T31 cases, and in east-central North America in the T21 case----may be due to the simplified experiment design and selected parameter values (some hints of this is seen in Fig.\,S2). However, it is important to stress that the ice evolution is ultimately controlled by the quality of the atmospheric forcing data, which we can show is strongly compromised at sufficiently coarse horizontal grids. Based on these results we conclude that a lower practical resolution bound for traditional climate-model experiments is likely to be somewhere around T31, and possibly somewhat higher (nominal T42 or even T85 resolution) for coupled climate--ice-sheet simulations.

\begin{acknowledgements}
We thank the editor Thomas M\"olg, two anonymous reviewers, and Irina Rogozhina and Raymond Sellevold for critically evaluating this manuscript.
We acknowledge B. Otto-Bliesner and J. Kleman and their collaborators for producing and making publicly available the CCSM3 LGM simulation and LGM ice-sheet reconstruction that were used as basis for our experiments.
The AGCM simulations were performed on resources provided by the Swedish National Infrastructure for Computing (SNIC) at the National Supercomputing Center (NSC) that is financially supported by Swedish Research Council (Vetenskapsr{\aa}det; VR). The ice-sheet model simulations were carried out on resources provided by LOEWE Frankfurt Centre for Scientific Computing (LOEWE-CSC).
%

This work was financially supported by the National Science Foundation (NSF) and the US Department of Energy (DOE).
\end{acknowledgements}

%%%%%%%%%%

\clearpage

\begin{table}[t]
\caption{
Resolution specific settings. The top two rows show the horizontal resolution in degrees
[$^\circ$] and in number of grid cells [lat$\times$lon], respectively.
The run cost (third row) is normalized with respect to the T21 case and estimates the number of numerical operations required to simulate one model year, based on the grid size and the nominal time step [s] used at each resolution (fourth row).
The horizontal biharmonic (fourth order) diffusion coefficient is given in units of $10^{15}$\,m$^4$s$^{-1}$ (bottom row). }
%
\label{table:cam3_settings}
{\begin{tabular}{rcccc}
\tophline
 & T85 & T42 & T31 & T21 \\
\middlehline
Resolution  & 1.4 & 2.8 & 3.8 & 5.6\\
%
Grid size & 128$\times$256 & 64$\times$128 & 48$\times$96 & 32$\times$64 \\
%
Run cost & 48 & 6 & 2.25 & 1 \\
%
Time step & 600 & 1200 & 1800 & 1800 \\
%
Diffusion & 1 & 10 & 20 & 20 \\
\bottomhline
\end{tabular}}
\belowtable{
}
%}
\end{table}

\clearpage

%% FIGURES

\begin{figure}[t]
  \includegraphics[width=1.\textwidth]{PSI_CLDTOT.pdf}
 \caption{
 (Left) Summer (JJA) 500 hPa eddy streamfunction [m$^2$s$^{-1}$] (shading; zonal mean removed) and zonal wind [m\,s$^{-1}$] (contours; 10\,m\,s$^{-1}$ intervals starting at 20\,m\,s$^{-1}$); (right) vertically integrated (total) cloudiness [$\%$].
 %
 The 500\,m ice-sheet topography from the LGM reconstruction is indicated by the heavy contours (interpolated to the different horizontal resolutions).
 }
 \label{fig:psi_cldtot}
\end{figure}

\clearpage

%%%%%%%%%%%%%%

\begin{figure}[t]
% \noindent\includegraphics[width=1.\textwidth]{TS.pdf}
  \includegraphics[width=1.\textwidth]{TS.pdf}
 \caption{
 Summer (JJA) surface temperature [$^\circ$C] from the different resolution atmospheric climatologies. The full fields are shown in the left panels (a,b,c,d), and the difference with respect to the T85 case is shown on the right (e,f,g).
 %
 The 500\,m ice-sheet topography from the LGM reconstruction is indicated by the heavy contours (interpolated to the different horizontal resolutions).
 }
 \label{fig:ts}
\end{figure}

\clearpage

%%%%%%%%%%%%%%%%%%%%%%%

\begin{figure}[t]
 \includegraphics[width=1.\textwidth]{CumPrec.pdf}
 \caption{
 Cumulative sum of precipitation (liquid + solid) over the year (total annual amount) [mm\,year$^{-1}$] from the different resolution atmospheric climatologies. The full fields are

shown in the left column (panels a,b,c,d), and the difference with respect to the T85 case is shown on the right (panels e,f,g).
  %
  The 500\,m ice-sheet topography from the LGM reconstruction is indicated by the heavy contours (interpolated to the different horizontal resolutions).
  }
  \label{fig:prect}
\end{figure}

\clearpage

%%%%%%%%%%%%%%%
\begin{figure}[t]
  \includegraphics[width=1.\textwidth]{ice.pdf}
  \caption{
  Equilibrium ice thickness [m] when using different ablation parameterizations in the surface mass balance scheme: (top) default method in SICOPOLIS; (middle) method by \citet{FaustoEA2009JGlac}; and (bottom) method by \citet{TarasovPeltier2002}. The atmospheric climatologies from the (a,e,i) T85; (b,f,j) T42; (c,g,k) T31; and (d,h,l) T21 resolution simulations were used, respectively.
  The 500\,m ice-sheet topography from the LGM reconstruction is indicated by the heavy contours (interpolated to the different horizontal resolutions).
  }
  \label{fig:ice}
\end{figure}